# Non-invasive plasma glycomic and metabolic biomarkers of post-treatment control of HIV

Leila B. Giron[1,10], Clovis S. Palmer[2,3,10], Qin Liu[1], Xiangfan Yin[1], Emmanouil Papasavvas[1], Radwa Sharaf[4], Behzad Etemad[4], Mohammad Damra[1], Aaron R. Goldman[1], Hsin-Yao Tang[1], Rowena Johnston[5], Karam Mounzer[6], Jay R. Kostman[6], Pablo Tebas[7], Alan Landay[8], Luis J. Montaner[1], Jeffrey M. Jacobson[9], Jonathan Z. Li[4] & Mohamed Abdel-Mohsen[1✉]

Non-invasive biomarkers that predict HIV remission after antiretroviral therapy (ART) interruption are urgently needed. Such biomarkers can improve the safety of analytic treatment interruption (ATI) and provide mechanistic insights into the host pathways involved in post-ART HIV control. Here we report plasma glycomic and metabolic signatures of time-to-viral-rebound and probability-of-viral-remission using samples from two independent cohorts. These samples include a large number of post-treatment controllers, a rare population demonstrating sustained virologic suppression after ART-cessation. These signatures remain significant after adjusting for key demographic and clinical confounders. We also report mechanistic links between some of these biomarkers and HIV latency reactivation and/or myeloid inflammation in vitro. Finally, machine learning algorithms, based on selected sets of these biomarkers, predict time-to-viral-rebound with 74% capacity and probability-of-viral-remission with 97.5% capacity. In summary, we report non-invasive plasma biomarkers, with potential functional significance, that predict both the duration and probability of HIV remission after treatment interruption.

[1] The Wistar Institute, Philadelphia, PA, USA. [2] The Burnet Institute, Melbourne, VIC, Australia. [3] Department of Infectious Diseases, Monash University, Melbourne, VIC, Australia. [4] Department of Medicine, Brigham and Women's Hospital, Harvard Medical School, Boston, MA, USA. [5] amfAR, The Foundation for AIDS Research, New York, NY, USA. [6] Philadelphia FIGHT, Philadelphia, PA, USA. [7] University of Pennsylvania, Philadelphia, PA, USA. [8] Rush University, Chicago, IL, USA. [9] Case Western Reserve University School of Medicine, Cleveland, OH, USA. [10] These authors contributed equally: Leila B. Giron, Clovis S. Palmer. ✉email: mmohsen@Wistar.org

  **1**

Several therapeutic strategies are being tested in clinical trials to reduce the size of HIV reservoirs to a point where virologic control can be achieved without antiretroviral therapy (ART)[1]. The success of these strategies depends on the capacity to determine if potential interventions have made a meaningful impact on the HIV reservoir, i.e., if they have extended the likely period of ART-free remission following treatment discontinuation. Because the impact of interventions on the total body burden of HIV cannot be measured with current technologies, HIV cure-focused clinical trials rely on an analytic treatment interruption (ATI) as the only definitive approach to evaluate the effectiveness of interventions[2]. However, this approach is costly, cumbersome, and poses some risk to both study participants and the community. These realities highlight the urgent need for biomarkers that can predict time-to-viral-rebound after treatment interruption and can be leveraged to guide clinical decision making. Such biomarkers could improve the safety of ATIs and accelerate the development of an HIV cure by providing a means for selecting only the most promising therapies for testing by ATIs[3]. These biomarkers also could provide mechanistic insights into the molecular and biochemical pathways involved in post-ART control of HIV.

In the last few years, a small number of immunophenotypic and virologic measurements have been associated with time-to-viral-rebound. For example, levels of exhaustion markers on CD4+ T cells, measured pre-ART, correlated with time-to-rebound[4]. However, when assessed during ART these measures fail as biomarkers[4]. Levels of cell-associated HIV DNA[5] and RNA[6,7], as well as features of plasmacytoid dendritic cells[8], during ART, correlate with viral rebound after ART cessation; however, the correlations are generally weak or modest. Thus, as of now, there are no sufficiently reliable or validated biomarkers that can be leveraged to guide clinical decision making.

Whereas the majority of HIV-infected individuals experience rapid viral rebound after ART interruption[6], a rare population of individuals, termed post-treatment controllers (PTCs), demonstrate sustained virologic suppression for several months to years after ART cessation[9]. The mechanisms underlying viral control in these individuals are not completely understood. Nonetheless, they represent a clinically relevant model for viral control post-ART[10]. The existence of individuals with this phenotype raises the question: is it possible to define a set of biomarkers that can predict the probability-of-viral-rebound after a potentially successful intervention (i.e., biomarkers that can predict the likelihood of achieving a PTC phenotype after ART cessation)? These biomarkers could also provide critical insights into the mechanisms that underlie this clinically relevant and desirable phenotype.

We have been taking advantage of work in the emerging fields of glycomics and metabolomics to identify robust, host-specific plasma biomarkers that can predict the duration and probability of viral remission after treatment interruption. Plasma glycoproteins (including antibodies; immunoglobulin G (IgGs)) and plasma metabolites enter the circulation from tissues through active secretion or leakage. Therefore, their levels and chemical characteristics can reflect the overall status of multiple organs, making them excellent candidates for biomarker discovery. Indeed, glycomic features in total plasma and on IgG have been identified as biomarkers for inflammatory bowel disease, cancer, and diabetes[11–13]. In addition, glycans on circulating glycoproteins have functional significance, as glycans play essential roles in mediating immunological functions, including antibody-dependent cell-mediated cytotoxicity (ADCC) and pro- and anti-inflammatory activities[14–16]. Similarly, plasma metabolites have been investigated as diagnostic and prognostic biomarkers in several diseases such as heart disease, and cancer[17–19]. Similar to plasma glycans, plasma metabolites are biologically active molecules that can regulate immunological responses, including inflammatory responses[20,21].

In a recent pilot study[22], we identified several plasma glycomic structures whose pre-ATI levels associate with delayed viral rebound after ART discontinuation. These were the digalactosylated glycans on bulk IgG, called G2, as well as fucose (total and core) and N-Acetylglucosamine (GlcNac) on total plasma glycoproteins[22]. However, that study only considered possible glycomic biomarkers, and was a small pilot that did not address the potentially confounding effects of age, sex, ethnicity, duration-on-ART, time of ART initiation (treatment at early vs. chronic stage of infection), or pre-ATI CD4 count.

In this current study, we first extended our biomarker discovery by performing metabolomic analyses on one of the two cohorts used in the pilot[22]. This was a cohort of 24 HIV-infected, ART-suppressed individuals who had participated in an open-ended ATI study without concurrent immunomodulatory agents. Our metabolomic analysis identified a select set of metabolites whose pre-ATI levels associate with time-to-viral-rebound. These metabolites belong to metabolic pathways known to impact inflammatory responses. We confirmed the direct, functional impact of some of these metabolites on latent HIV reactivation and/or macrophage inflammation in vitro. We then profiled both the plasma glycome and metabolome of a large cohort of 74 HIV-infected, ART-suppressed individuals who underwent ATI during several AIDS Clinical Trials Group (ACTG) clinical trials. This cohort contains 27 PTCs and 47 post-treatment non-controllers (NCs). Using this cohort, we confirmed that a set of plasma glycans and metabolites were able to predict time-to-viral-rebound and probability-of-viral-rebound, even after adjusting for several potential demographic and clinical confounders. Finally, using machine-learning models, we combined this set of biomarkers into two multivariate models: a model that predicts time-to-viral-rebound with 74% capacity; and a model that predicts probability-of-viral-rebound with 97.5% capacity. Together, we identified non-invasive plasma biomarkers, with potential functional significance, that predict duration and probability of viral remission after treatment interruption.

## Results

**Characteristics of study cohorts.** In this study, we employed two ATI cohorts: (1) The Philadelphia cohort: a group of 24 HIV-infected individuals on suppressive ART who underwent an open-ended ATI[22,23]. This cohort had a wide distribution of viral rebound times (14–119 days; median=28; Supplementary Table 1)[22]. Importantly, this cohort underwent ATI without concurrent immunomodulatory agents that might confound our signatures at the initial phase of the study[22,23]. (2) The ACTG cohort: a cohort that combined 74 participants from six ACTG ATI studies (ACTG 371[24], A5024[25], A5068[26], A5170[27], A5187[28], and A5197[29]), which tested the efficacy of different HIV vaccines and/or immunotherapies. These six ATI studies included 600 participants and identified 27 PTCs among their participants. Our ACTG cohort included all 27 PTCs and 47 matched non-controllers (NCs) from the same studies. The definition of post-treatment control was: remaining off ART for ≥24 weeks post-ATI with viral load (VL) ≤ 400 copies for at least 2/3 of time points; no ART in the plasma; and no evidence of spontaneous control pre-ART. The 47 NCs rebounded before meeting PTC criteria[30,31], The PTC and NC groups within the ACTG cohort are matched for sex, age, ethnicity, percent treated during early infection, ART duration, and pre-ATI CD4 count (Table 1, Supplementary Fig. 1, and Supplementary Table 2). Notably, the individuals in our

**Table 1 Demographic and clinical characteristics of PTCs and NCs from the ACTG cohort.**

|  | PTCs (*N* = 27) | NCs (*N* = 47) |
|---|---|---|
| Male, *n* (%) | 21 (78) | 40 (85) |
| Age, years, median (IQR) | 41 (8) | 41 (10) |
| Early treated, *n* (%) | 10 (37) | 19 (40) |
| Years on ART, median (IQR) | 4.2 (4.7) | 3.4 (4.5) |
| Pre-ATI CD4 count (cells/mm$^3$), median (IQR) | 885 (224) | 846 (301.5) |
| Days to VL ≥ 1000 copies/ml, median (IQR) | 111 (282) | 27 (25.5) |
| Days to two consecutive VL ≥ 1000 copies/ml, median (IQR) | 331 (278) | 27 (27) |
| Ethnicity |  |  |
| Caucasian, *n* (%) | 17 (63) | 30 (64) |
| African American, *n* (%) | 7 (26) | 9 (19) |
| Hispanic, *n* (%) | 3 (11) | 8 (17) |

*PTCs* post-treatment controllers, *NCs* post-treatment non-controllers, *ART* antiretroviral therapy, *IQR* interquartile range, *VL* viral load.

ACTG cohort had participated in clinical trials where they had received, or not, different HIV vaccines and/or immunotherapies[24–29]. This important feature of this cohort allows for identifying/validating markers that predict duration and probability of viral remission independent of potential interventions. Metabolic and glycomic analyses were performed on samples collected at one timepoint, shortly before the ATI.

**Elevated pre-ATI levels of plasma markers of glutamate and bile acid metabolism associate with a delayed viral rebound in the Philadelphia Cohort.** We first aimed to examine the utility of plasma metabolites as biomarkers of time-to-HIV-rebound after ART cessation. Towards this goal, we measured levels of plasma metabolites from the Philadelphia cohort[22,23]. Using an untargeted mass spectrometry (MS)-based metabolomics analysis, we identified a total of 179 metabolites in the plasma samples collected immediately before the ATI. Then, we applied the Cox proportional-hazards model to identify metabolomic signatures of time-to-viral-rebound. As shown in Fig. 1a, higher pre-ATI levels of 13 plasma metabolites were significantly associated with a longer time-to-viral-rebound with $P < 0.05$ and false discovery rate (FDR) < 20%. In contrast, higher pre-ATI levels of 12 plasma metabolites were significantly associated with a shorter time-to-viral-rebound. When participants were separated into low or high groups, based on the median of each of the 25 metabolic markers, the pre-ATI levels of 20 of 25 metabolites significantly indicated hazards of viral-rebound over time using the Mantel–Cox test (Fig. 1b and Supplementary Table 3).

We next sought to determine if the 25 metabolites that associated with time-to-viral-rebound shared similar metabolic pathways. Multi-analysis combining KEGG and the STRING Interaction Network (focusing on metabolite-associated enzymatic interactions) revealed that most of the 13 metabolites whose pre-ATI levels associated with a longer time-to-viral-rebound belonged to two major metabolic pathways. Specifically, five metabolites lay within the anti-inflammatory glutamate/ tricarboxylic acid (TCA) cycle pathway, and three were intermediates within the primary bile acid biosynthesis pathway (Fig. 1c). Confirmatory analysis on these 13 metabolites using the MetaboAnalyst 3.0 pathway feature (http://www.metaboanalyst. ca/) showed enrichment in glutamate metabolism ($P = 0.00068$) and the bile acid biosynthesis pathway ($P = 0.0399$) (Fig. 1c and Supplementary Table 4).

**Elevated pre-ATI levels of plasma markers of pyruvate and tryptophan metabolism associate with an accelerated viral rebound in the Philadelphia Cohort.** Multi-analysis of the 12 metabolites whose pre-ATI levels associated with shorter time-to-viral-rebound showed four intermediates in the tryptophan metabolism pathway and three that are central players in the pro-inflammatory pyruvate pathway (Fig. 1d). These observations were confirmed for the 12 metabolites using MetaboAnalyst 3.0, which demonstrated enrichment in pyruvate metabolism ($P = 0.0065$) (Fig. 1d and Supplementary Table 4). The roles of key discovered metabolites within the glutamate, bile acids, tryptophan, and pyruvate pathways are graphically illustrated in Supplementary Fig. 2. These data reveal a previously undiscovered class of plasma metabolic biomarkers that are associated with time-to-viral rebound post-ATI. They further demonstrate that these biomarkers belong to a specific set of metabolic pathways that may play a previously unrecognized role in HIV control.

**L-glutamic acid and pyruvate modulate latent HIV reactivation and/or macrophage inflammation in vitro.** Among the top candidate metabolic biomarkers from Fig. 1 are L-glutamic acid (glutamate metabolism) and pyruvic acid (pyruvate metabolism). The higher pre-ATI levels of L-glutamic acid and pyruvic acid were associated with a longer or a shorter time-to-viral-rebound, respectively. These two metabolites can impact inflammation in opposing directions. Glutamate controls the anti-inflammatory TCA cycle through its conversion by glutamate dehydrogenase to α-ketoglutarate[32,33], whereas pyruvate is centrally positioned within the pro-inflammatory glycolytic pathway[34,35]. We therefore sought to determine if these two metabolites had a direct, functional impact on latent HIV transcription and/or myeloid inflammation.

We first assessed the impact of each of these two metabolites on latent HIV reactivation using the established "J-Lat" model of HIV latency. J-Lat cells harbor a latent, transcriptionally competent HIV provirus that encodes green fluorescent protein (GFP) as an indicator of reactivation (Fig. 2a)[36]. There are several clones of the J-Lat model with different characteristics, including the type of stimulation to which they respond. For example, the 5A8 clone is the only J-Lat clone responsive to αCD3/αCD28 stimulation. We examined the impact of L-glutamic acid and pyruvate on latent HIV reactivation using two J-Lat clones, 5A8 and 10.6. L-glutamic acid significantly inhibited the ability of phorbol-12-myristate-13-acetate (PMA)/ionomycin or αCD3/αCD28 to reactivate latent HIV in clone 5A8 without impacting viability, compared to stimuli alone controls (Fig. 2b). L-glutamic acid also inhibited the ability of PMA/ionomycin or TNFα to reactivate latent HIV in clone 10.6 without impacting viability, compared to stimuli alone controls (Fig. 2c). Finally, to test the impact of glutamine in the RPMI media on our results, J-Lat 5A8 cells were cultured in glutamine-free media and treated with L-glutamic acid at different doses, in the presence or absence of PMA/I. As shown in Supplementary Fig. 3, L-glutamic acid inhibited the ability of PMA/I to reactivate the 5A8 J-Lat clone in a dose-dependent manner in glutamine-free media. These data demonstrate that a plasma metabolite, L-glutamic acid, can inhibit latent viral reactivation, consistent with the observation that pre-ATI levels of L-glutamic acid predicted a longer time-to-viral-rebound.

Beyond the direct impact on latent viral reactivation, plasma metabolites may exert effects on myeloid inflammation, and such effects may underlie HIV control during ATI. This possibility was tested by examining the effects of L-glutamic acid and pyruvate on

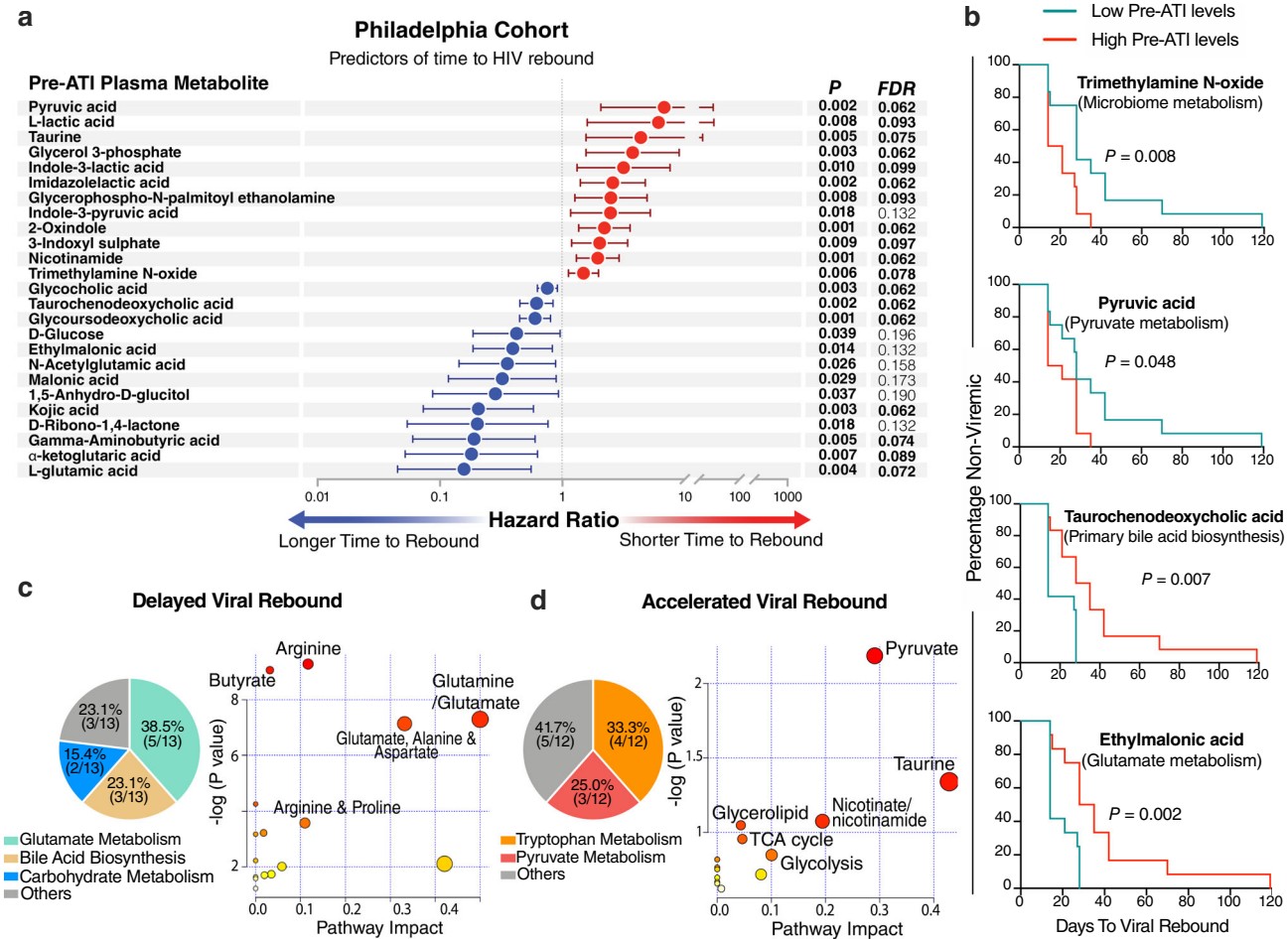

**Fig. 1 Plasma metabolites associate with time-to-viral-rebound in the Philadelphia Cohort. a** Cox proportional-hazards model of metabolites associated with a longer (blue) or a shorter (red) time-to-viral rebound during Analytic Treatment Interruption (ATI). $n = 24$ biologically independent samples. Data are presented as hazard ratios with 95% confidence intervals. Two-sided $P$ value of each independent variable in the model was used. False Discovery Rate (FDR) was calculated using Benjamini–Hochberg method to correct for multiple comparisons. **b** Two-sided Mantel–Cox test analysis of four selected metabolites from **a**. Low pre-ATI levels = lower than group median; High pre-ATI levels = higher than group median. $n = 24$ biologically independent samples. **c** Pathway analysis of the 13 metabolites (blue circles in **a**) whose pre-ATI levels are associated with a delayed viral rebound. Left image: a multi-analysis approach combining KEGG and STRING Interaction Network. Right image: unbiased analysis using MetaboAnalyst 3.0 (http://www.metaboanalyst.ca/) where the node color is based on $P$ value, and the node radius is based on the pathway impact value. The pathway impact is determined by normalizing the sum of matched metabolites to the sum of all metabolites in each pathway. **d** Pathway analysis of the 12 metabolites (red circles in **a**) whose pre-ATI levels are associated with an accelerated viral rebound. Analysis was performed as in panel (**c**). Source data are provided as a Source Data file.

lipopolysaccharides (LPS)-mediated secretion of pro-inflammatory cytokines from THP-1 derived macrophage-like cells. These cells are characterized by high basal glycolytic activity, which closely reflects the Warburg-like phenotype observed in HIV-infected individuals[37]. They also exhibit similar inflammatory responses as primary cells under similar in vitro conditions[38]. Cells were treated with L-glutamic acid, pyruvate, or appropriate controls for 2 h before stimulating with LPS and IFNγ for 5 h (Fig. 2d). L-glutamic acid inhibited LPS/IFNγ-mediated production of pro-inflammatory cytokines such as IL-6 and TNFα (Fig. 2e; other cytokines are shown in Supplementary Fig. 4a). Consistently, L-glutamic acid also increased production of anti-inflammatory IL-10 (Fig. 2e). Conversely, pyruvate increased IL-6 and TNFα secretion (Fig. 2f; other cytokines are shown in Supplementary Fig. 4b). These data demonstrate not only that some metabolites associate with time-to-viral-rebound, but also that there is a plausible, functionally significant link between these metabolic biomarkers and viral control during and following ATI.

**Pre-ATI plasma glycomic and metabolic biomarkers associate with time-to-viral-rebound in the ACTG Cohort.** Our recent pilot study showed that pre-ATI levels of a specific set of glycans predicted a longer time-to-viral rebound after ART discontinuation[22]. However, this small pilot study did not correct for confounders such as age, sex, and nadir CD4 count on viral rebound. We hypothesized that a set of plasma glycans and metabolites we identified in that pilot study[22], as well as in the results shown in Fig. 1, can predict time-to-viral-rebound and/or probability-of-viral-rebound, even after adjusting for potential demographic and clinical confounders. For this analysis of a larger validation cohort, we analyzed samples from the ACTG cohort.

We analyzed the plasma metabolome of samples collected from this cohort before ATI. A total of 226 metabolites were identified using MS-based metabolomics analysis. In addition, we analyzed the plasma glycome of the same samples by applying two different glycomic technologies. First, we used capillary electrophoresis to identify the N-linked glycans of total plasma

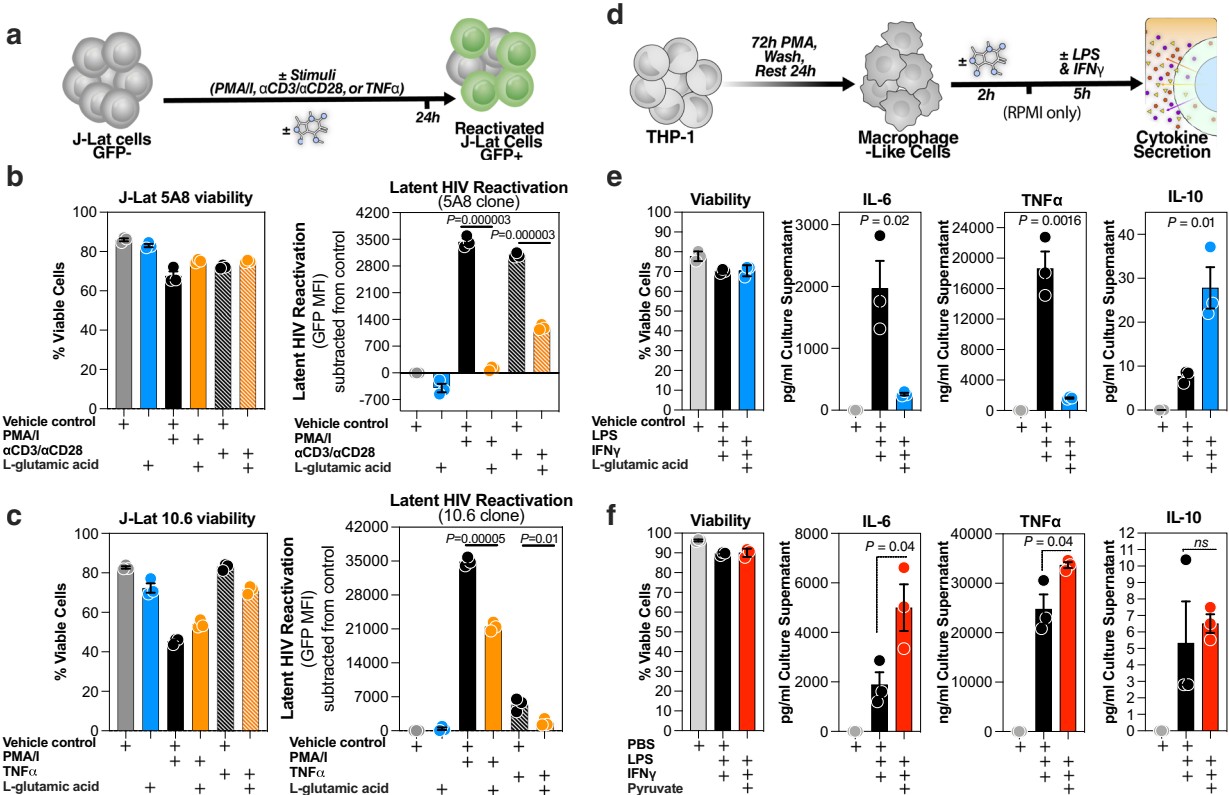

**Fig. 2 L-glutamic acid and pyruvate directly impact latent HIV reactivation and/or macrophage inflammation. a** JLat 5A8 or 10.6 clones were stimulated with appropriate stimuli in the presence or absence of L-glutamic acid or vehicle control (cell culture suitable HCl solution). Geometric mean fluorescence intensity (MFI) of HIV-regulated green fluorescent protein (GFP) expression was measured by flow cytometry. Cell viability was determined by LIVE/DEAD aqua staining. **b** J-Lat 5A8 cells ($n = 3$ independent experiments) and (**c**) J-Lat 10.6 cells ($n = 3$ independent experiments), were treated with PMA/I (16 nM/500 nM), ImmunoCult Human CD3/CD28 T Cell Activator (25 μl per $10^6$ cells), or TNFα (10 ng/ml) in the presence or absence of L-glutamic acid (5 mM) or appropriate control. Bar graphs display mean ± SD values, and statistical comparisons were performed using two-tailed unpaired t-tests. **d** THP-1 cells ($n = 3$ independent experiments) were differentiated into macrophage-like cells using PMA. Cells were then treated with L-glutamic acid (5 mM), pyruvate (2 mM), or appropriate controls for 2 h prior to LPS/IFNγ stimulation for 5 h. Cell viability was determined by LIVE/DEAD aqua staining, and cytokine secretion was measured in the supernatants using ELISA and MSD platform multiplex assay (**e**) L-glutamic acid significantly inhibited LPS/IFNγ-mediated secretion of pro-inflammatory cytokines such as IL-6 and TNFα but significantly increased the anti-inflammatory IL-10 release. Bar graphs display mean ± SD, and statistical comparisons were performed using two-tailed unpaired t-tests. **f** Pyruvate significantly increased LPS/IFNγ-mediated secretion of IL-6 and TNFα. Bar graphs display mean ± SD, and statistical comparisons were performed using two-tailed unpaired t-tests. Source data are provided as a Source Data file.

glycoproteins (we identified 24 glycan structures, their names and structures are listed in Supplementary Fig. 5) and of isolated plasma IgG (we identified 22 glycan structures, their names and structures are listed in Supplementary Fig. 6). Second, we used a 45-plex lectin microarray to identify total (N and O linked) glycans on plasma glycoproteins. The lectin microarray enables sensitive identification of multiple glycan structures by employing a panel of 45 immobilized lectins (glycan-binding proteins) with known glycan-binding specificity, resulting in a "glycan signature" for each sample (the 45 lectins and their glycan-binding specificities are listed in Supplementary Table 5)[39].

We used the Cox proportional-hazards model and a set of highly stringent criteria to identify sets of glycans or metabolites whose pre-ATI levels associate with either time to VL ≥ 1000 copies/ml (Fig. 3 top panel) or time to two consecutive VL ≥ 1000 copies/ml (Fig. 3 bottom panel). To ensure high stringency, we only considered markers with a hazard ratio (HR) ≥ 2 or ≤0.5. We also only included markers with FDR < 10%, or markers that had emerged from the Philadelphia cohort (Fig. 1 and our previous pilot study[22]). Importantly, we only included markers that remained significant ($P < 0.05$) after adjusting for age, sex, ethnicity, ART initiation (during early or chronic HIV infection),

ART duration, or pre-ATI CD4 count (Supplementary Table 6). These combined strict criteria identified a plasma signature that predicted shorter time-to-rebound to VL ≥ 1000 copies/ml, comprising four glycan structures and one metabolite (Fig. 3 top panel, red). These five markers include the highly sialylated plasma N-glycan structure A3G3S3, GalNAc-containing glycans (also known as T-antigen; measured by binding to both MPA and ACA lectins), and the metabolite pyruvic acid. We also identified a signature that associated with a longer time-to-rebound to VL ≥ 1000 copies/ml, comprising seven glycan structures and one metabolite, notably the digalactosylated G2 glycan structure on plasma bulk IgG, fucosylated glycans in plasma (binding to AAL lectin), GlcNac glycans in plasma (binding to DSA, UDA, and STL lectins), and the metabolite L-glutamic acid (Fig. 3 top panel, blue).

Turning to markers that associated with time to two consecutive VL ≥ 1000 copies/ml, and applying the same strict criteria, we identified five glycomic markers whose pre-ATI levels associate with shorter time-to-rebound post-ATI, including A3G3S3 in plasma and T/Tn-antigens (binding to MPA, ACA, and ABA lectins) (Fig. 3 bottom panel, red). We also identified seven glycan structures and two metabolites whose pre-ATI levels

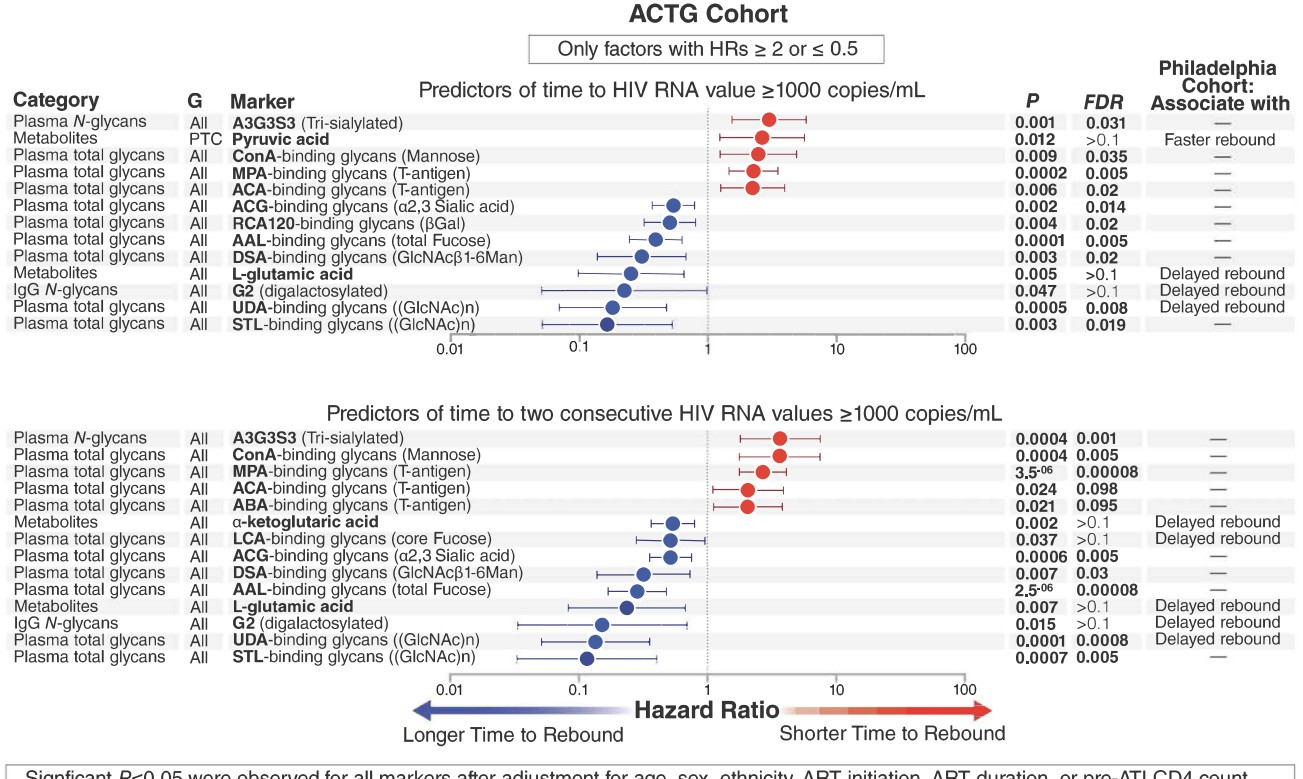

**Fig. 3 Hazard ratios of plasma glycomic and metabolic markers that associated with time-to-viral-rebound in the ACTG Cohort.** Cox proportional-hazards model of glycomic and metabolic markers of time to (top panel) VL ≥ 1000 copies/ml or (bottom panel) two constitutive VL ≥ 1000 copies/ml within the ACTG Cohort. Data are presented as hazard ratios with 95% confidence intervals. Two-sided *P* value of each independent variable in the model was used. False Discovery Rate (FDR) was calculated using Benjamini–Hochberg method to correct for multiple comparisons. *n* = 74 biologically independent samples. G = group (All = using data from all 74 participants and PTC = using data from only the 27 PTCs within the ACTG Cohort). HRs = hazard ratios. Source data are provided as a Source Data file.

predicted a longer time-to-rebound, including G2 glycan structure on bulk IgG, core fucosylated glycans (binding to LCA lectin) in plasma, total fucosylated glycans (binding to AAL lectin) in plasma, GlcNac glycans (binding to DSA, UDA, and STL lectins) in plasma, and the metabolites oxoglutaric acid (α-ketoglutaric acid) and L-glutamic acid (Fig. 3 bottom panel, blue). The significance of several of these markers was also confirmed using the Mantel–Cox test in an independent analysis (Fig. 4). In sum, using stringent analysis criteria that also took into account potential confounders, we identified plasma glycomic/metabolomic signatures of time-to-viral-rebound after ART discontinuation in this independent, heterogeneous cohort of individuals who underwent ATI and received or not several different interventions before ATI.

**Levels of pre-ATI plasma glycomic and metabolic markers that associate with time-to-viral-rebound are linked to levels of total, intact, and defective cell-associated HIV DNA as well as cell-associated HIV RNA in the blood.** We next examined whether the plasma glycans and metabolites (Fig. 3) that associated with time-to-viral-rebound also reflected levels of virological markers of HIV persistence. We measured levels of peripheral blood mononuclear cell (PBMC)-associated total HIV DNA and HIV RNA by qPCR on a subset of 32 individuals from the ACTG cohort. Pre-ATI levels of cell-associated HIV DNA and RNA have been shown to correlate with time-to-viral-rebound in several previous studies[5–7]. Indeed, in our cohort, levels of cell-associated HIV DNA and RNA were lower in PTCs compared to NCs and predicted time-to-viral-rebound using the

Cox proportional-hazards model (Supplementary Fig. 7). When we examined the associations between these measures and our glycomic and metabolic markers, we found that pre-ATI levels of total fucose (binding to AAL lectin), which predicted delayed viral rebound, showed a significant inverse correlation with pre-ATI levels of cell-associated HIV DNA and RNA (Fig. 5a–c). Similarly, pre-ATI levels of core fucose (binding to LCA lectin), which also predicted delayed viral rebound, also showed an inverse correlation with pre-ATI levels of cell-associated HIV DNA and RNA (Fig. 5a). Furthermore, total levels of (GlcNAc)n (binding to UDA and STL lectins), which predicted delayed viral rebound, had an inverse correlation with levels of total HIV DNA (Fig. 5a). Noteworthy, levels of pyruvic acid, whose pre-ATI levels predicted accelerated viral rebound, had a significant positive correlation with pre-ATI levels of cell-associated HIV DNA (Fig. 5a, d).

The majority of HIV DNA harbor mutations and/or deletions, rendering them defective[40]. Intact HIV proviruses can support viral transcription and translation; however, recently, it was also shown that some defective HIV proviruses can express viral RNA and proteins[41,42]. We sought to examine the potential links between our plasma markers and levels of intact and defective HIV DNA. To do this, we took advantage of near-full length sequencing data that were recently generated on a subset of 19 individuals from this cohort (10 PTCs and 9 NCs)[31]. Within these 19 individuals, levels of intact, defective, and hypermutated HIV DNA were lower in PTCs compared to NCs, and levels of defective HIV DNA predicted time-to-viral-rebound using the Cox proportional-hazards model (Supplementary Fig. 7). When

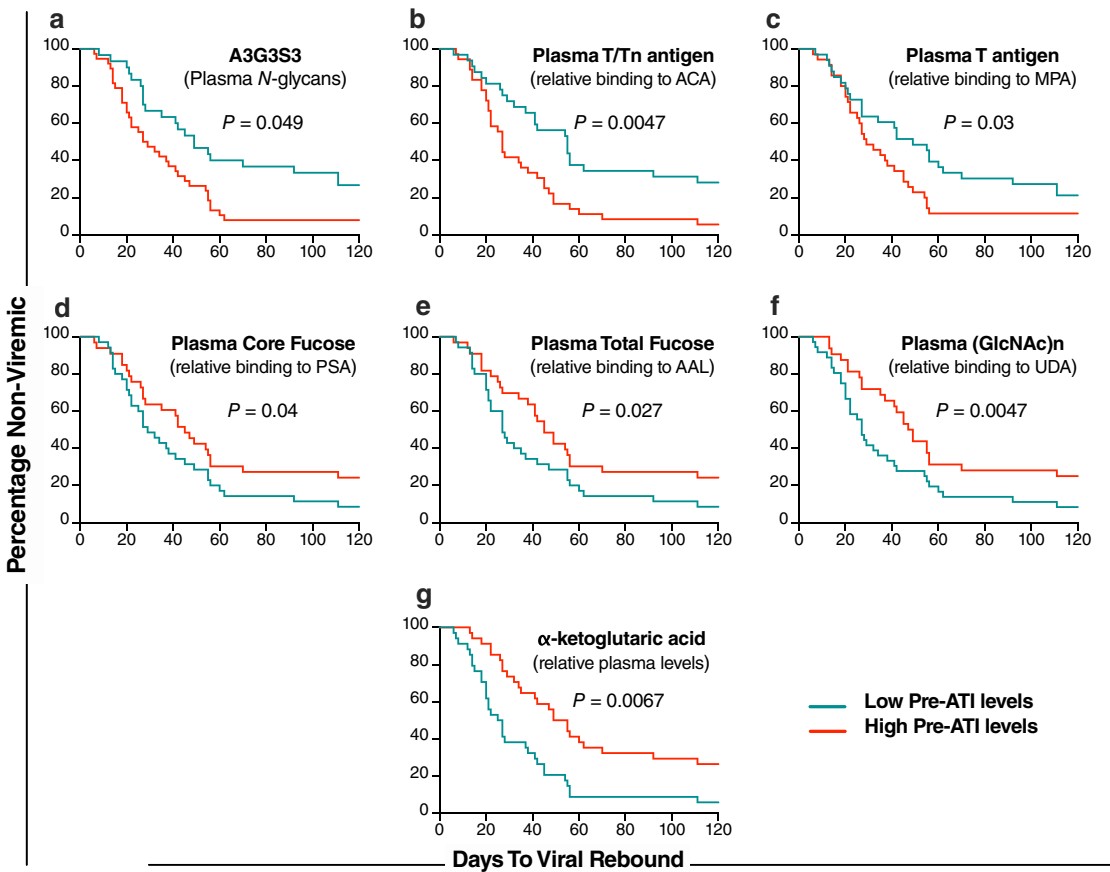

**Fig. 4 Mantel–Cox plots of plasma glycomic and metabolic markers that associated with time-to-viral-rebound in the ACTG Cohort.** Graphic representation of two-sided Mantel–Cox test illustrating six glycans (**a–f**) and one metabolite (**g**) that predicted time-to-viral-rebound in Fig. 3. Nominal *P* values are reported. Low pre-ATI levels = lower than the median; high pre-ATI levels = higher than the median. *n* = 74 biologically independent samples. Source data are provided as a Source Data file.

we examined the associations between these measures and our glycomic and metabolic markers, we found that pre-ATI levels of L-glutamic acid, which predicted delayed viral rebound in both the Philadelphia and ACTG cohorts, showed a significant inverse correlation with pre-ATI levels of intact and defective (but not hypermutated) cell-associated HIV DNA (Fig. 5a, e, f). In general, there was a trend for negative correlations between the pre-ATI levels of markers associated with a delayed viral rebound and the size of HIV reservoir (Fig. 5a). On the other hand, there was a trend for positive correlations between the pre-ATI levels of markers associated with a faster viral rebound and the size of HIV reservoir (Fig. 5a). These data provide more support for plausible connections between our discovered plasma markers and HIV persistence and control during ATI.

**Multivariable Cox model, using Lasso technique with cross-validation (CV), selected variables whose combination predicts time-to-viral-rebound.** As a single marker would be highly unlikely to strongly predict these complex virological milestones, we next sought to apply a machine-learning algorithm to identify a smaller set of plasma biomarkers (from Fig. 3) that together can predict time to VL ≥ 1000 copies/ml better than any of these biomarkers individually. The analysis considered biomarkers, both metabolites and/or glycan structures, that emerged as significant from the ACTG cohort (Fig. 3) and used data from ACTG samples with complete data sets (*n* = 70; four samples did not have a complete dataset). The machine-learning algorithm, Lasso (least absolute shrinkage and selection operator) regularization, selected

seven markers from the 13 that associated with time to VL ≥ 1000 copies/ml (Fig. 3 top panel), whose predictive values are independent, and which, when combined, enhance the predictive ability of the signature compared to any marker alone (Supplementary Table 7). Indeed, with the complete data from ACTG samples, a multivariable Cox regression model using these seven variables showed a concordance index (C-index) value of 74% (95% confidence interval: 68–80%), which is significantly higher than the C-index values obtained from Cox models using each variable individually (*P* < 0.05; Supplementary Table 7). Notably, these seven markers included four whose pre-ATI levels associated with accelerated rebound: A3G3S3, T-antigen (MPA and ACA lectins binding), and the metabolite pyruvic acid. The other three markers associated with delayed rebound: total fucose (AAL lectin binding), (GlcNAc)n (STL lectin binding), and the metabolite L-glutamic acid (Supplementary Table 7). To be conservative, a more robust estimate of the model's performance was calculated using a 5-fold cross-validated model in the ACTG cohort, which resulted in an average C-index of 70.6% with a variance of 0.0004 (Supplementary Table 8). Together, these data suggest that the multivariable model of combined plasma glycans and metabolites markers warrant further exploration for its capacity to predict time-to-viral-rebound in different settings.

**Pre-ATI plasma glycomic and metabolic markers distinguish PTCs from non-controllers (NCs).** Examining the glycan structures and metabolites obtained from the ACTG cohort, we identified eight glycan structures whose pre-ATI levels were

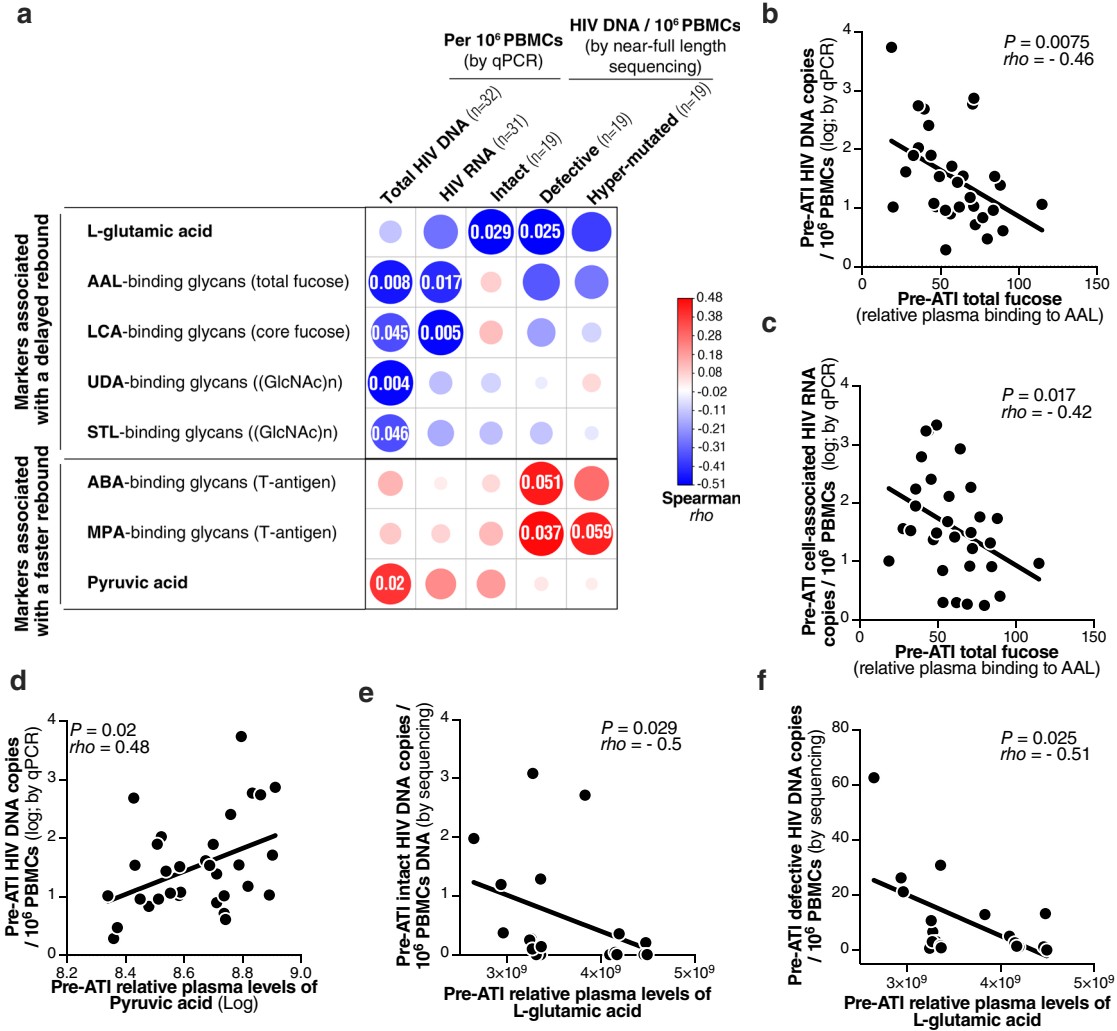

**Fig. 5 Plasma glycomic and metabolic markers of time-to-viral-rebound associate with levels of PBMC-associated HIV DNA (total, intact, and defective) and RNA in the ACTG Cohort. a** Two-sided Spearman's correlation heat-map showing associations between markers associated with time-to-viral-rebound (in rows) and levels of cell-associated HIV DNA and RNA (measured by qPCR) or levels of intact, defective, and hypermutated HIV DNA (measured by near-full length sequencing) (in columns). The size and color of circles represent the strength of the correlation, with blue shades represent negative correlations and red shades represent positive correlations. Numbers inside the circles are nominal $P$ values. **b**, **c** Inverse associations between pre-ATI plasma levels of total fucose and levels of pre-ATI cell-associated (**b**) HIV DNA or (**c**) HIV RNA. **d** Positive association between pre-ATI plasma levels of the metabolite pyruvic acid and levels of cell-associated HIV DNA. **e**, **f** Inverse associations between pre-ATI plasma levels of the metabolite L-glutamic acid and levels of intact (**e**) or defective (**f**) HIV DNA. All correlations were done using two-sides Spearman's rank correlation coefficient tests. For all panels, $n = 32, 31, 19, 19$, and 19 biologically independent samples were used for correlations with total HIV DNA (by qPCR), cell-associated HIV RNA (by qPCR), intact HIV DNA (by sequencing), defective HIV DNA (by sequencing), and hypermutated HIV DNA (by sequencing), respectively. Source data are provided as a Source Data file.

significantly different between PTCs and NCs with FDR < 0.1 (Fig. 6a–h). Among these eight glycans structures, three exhibited lower levels in the plasma of PTCs compared to NCs (FDR < 0.02), including the disialylated glycans, A2, in total IgG glycans; the highly sialylated glycans, A3G3S3, in plasma N-glycans; and T-antigen (binding to ABA lectin) (Fig. 6a-c); and five glycans were higher in PTCs compared to NCs (FDR ≤ 0.035). These were total fucose (binding to AAL lectin), core fucose (binding to LCA and PSA lectins), and (GlcNac)n (binding to STL and UDA lectins (Fig. 6d–h).

Examining metabolites, we found that pre-ATI levels of α-ketoglutaric acid and L-glutamic acid, both of which predicted delayed viral rebound, were higher in the plasma of PTCs compared to NCs ($P < 0.01$, Fig. 6i, j). Importantly, this set of 10 markers contains only those markers whose levels remained different ($P < 0.05$) between PTCs and NCs after adjusting for

age, sex, ethnicity, ART initiation, ART duration, or pre-ATI CD4 count (Supplementary Table 9). Together, these data suggest that a selective set of plasma glycans and metabolites can distinguish PTCs from NCs and may be used to predict the probability of viral rebound (i.e., the likelihood of PTC phenotype after ATI).

**Multivariable logistic model, using CV Lasso technique, selected variables whose combination predicts risk of viral rebound.** We next applied the Lasso regularization to select, from among the ten markers in Fig. 6, a set of markers whose combined predictive utility is better than the predictive utility of any of these ten markers individually. The analysis used biomarkers that emerged as significant from the ACTG cohort (Fig. 6) and only those samples with complete data sets ($n = 70$). Lasso selected seven markers from the ten identified as able to

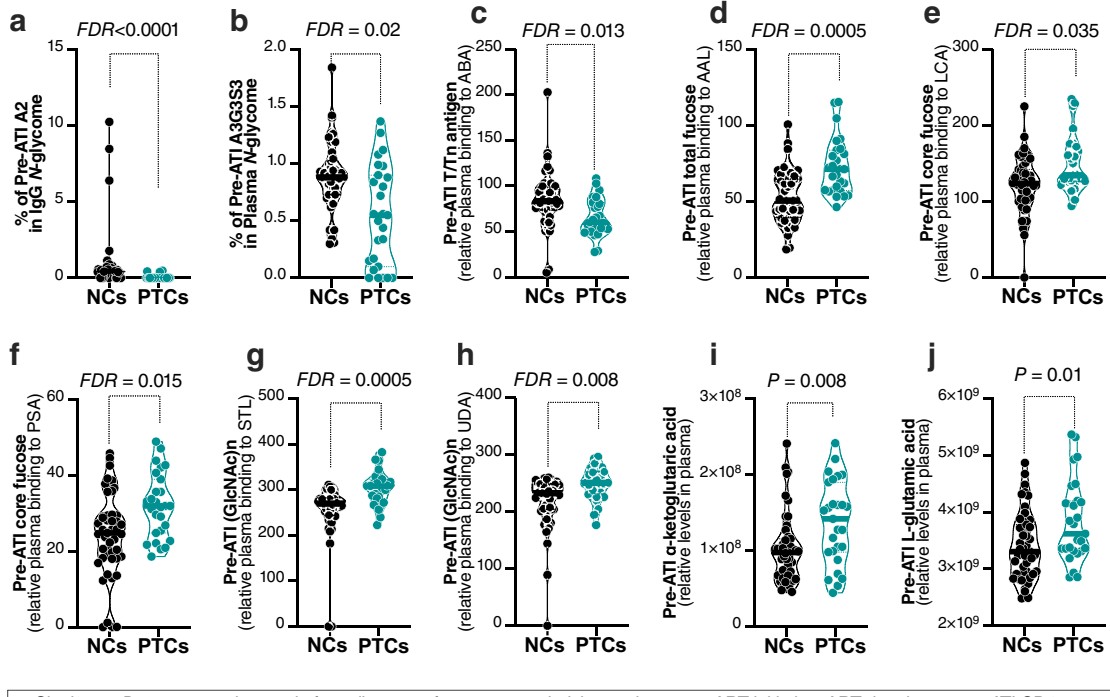

Significant *P*<0.05 were observed after adjustment for age, sex, ethnicity, study source, ART initiation, ART duration, or pre-ATI CD4 count

**Fig. 6 Plasma glycomic and metabolic markers that distinguish post-treatment controllers (PTCs) from non-controllers (NCs).** Pre-ATI levels of three glycan structures are lower in PTCs compared to NCs: (**a**) the disialylated glycans, A2, in the IgG glycome, (**b**) the highly sialylated glycan structure (A3G3S3), and (**c**) T/Tn antigen (measured as binding to ABA lectin). Pre-ATI levels of four glycan structures were higher in PTCs compared to NCs: (**d**) total fucose (binding to AAL lectin) in plasma, (**e**, **f**) core fucose (binding to LCA and PSA lectins) in plasma, and (**g**, **h**) (GlcNAc)n (binding to STL and UDA lectins). Pre-ATI levels of two metabolites were higher in PTCs compared to NCs: (**i**) α-ketoglutaric acid and (**j**) L-glutamic acid. All statistical comparisons were performed using a two-sided Mann–Whitney test. Truncated violin plots showing median. False Discovery Rate (FDR) was calculated using Benjamini–Hochberg (BH) method to correct for multiple comparison in panels (**a**–**f**) and nominal *P* values are reported in panels (**g**, **h**). Source data are provided as a Source Data file.

distinguish PTCs from NCs whose predictive values are independent and combing them enhances the predictive ability of the signature compared to each of these markers alone (Supplementary Table 10). Indeed, a multivariable logistic regression model using these seven variables showed an area under the ROC curve (AUC) value of 97.5% (Fig. 7a; 95% confidence interval: 94–100%), which is significantly higher than the AUC values obtained from logistic models using each variable individually ($P < 0.05$; Supplementary Table 10). A more robust estimate of the model's performance with a 5-fold cross-validated model in the ACTG cohort shows an average AUC of 94.7% with a variance of 0.0049 (Supplementary Table 8). These seven markers included three whose pre-ATI levels are lower in PTCs compared to NCs, namely A2, A3G3S3, and T-antigen (ABA lectin binding), and four whose pre-ATI levels were higher in PTCs compared to NCs, namely total fucose (AAL lectin binding), core fucose (LCA lectin binding), (GlcNAc)n (STL lectin binding), and the metabolite L-glutamic acid (Supplementary Table 10).

Next, a risk score predicting NC was estimated for each individual using the multivariable logistic model. We then examined the ability of these risk scores to classify PTCs and NCs from the ACTG cohort. As shown in Fig. 7b, the model correctly classified 97.7% of NCs (sensitivity) and 85.2% of PTCs (specificity) with an overall accuracy of 92.9%. This analysis highlights the potential utility of this risk score, estimated from the multivariable model and combining six plasma glycans and one metabolite, to predict the risk of NC post-ATI. This prediction can be used to select for ATI studies the individuals who are likely to achieve the PTC phenotype during HIV cure-focused clinical trials. In addition, the markers that are included

in this model might also serve as windows into the mechanisms that contribute to the PTC phenotype.

**Discussion**

In this study, we identified pre-ATI plasma glycomic and metabolomic biomarkers of both duration and probability of viral remission after treatment interruption. We observed a significant overlap between plasma markers that predicted time-to-viral rebound and markers that predicted the probability of viral rebound (i.e., predicted the PTC phenotype in comparison to the NC phenotype). Specifically, pre-ATI plasma levels of the anti-inflammatory L-glutamic acid, *N*-acetylglucosamine (GlcNac), and fucose were associated with both a delayed rebound and a higher likelihood to achieve viral remission. In contrast, pre-ATI plasma levels of the highly sialylated A3G3S3 and GalNAc-containing glycans (T/Tn-antigens) were associated with both an accelerated rebound and a lower likelihood of achieving viral remission. Notable differences included the digalactosylated G2 glycan on IgG glycome, whose pre-ATI levels were associated with longer time-to-viral-rebound but not the probability of viral rebound; and the disialylated IgG glycan, A2, whose pre-ATI levels associated with a higher probability of viral rebound but not with time-to-viral-rebound.

It is not surprising that a single marker cannot highly predict these complicated virological milestones (time to and probability of viral rebound). Therefore, we applied machine-learning algorithms to select the smallest number of variables that, when combined, maximizes the predictive utility of our signatures. The variables selected by the CV Lasso technique, when used in multivariate models, were able to predict time-to-viral rebound

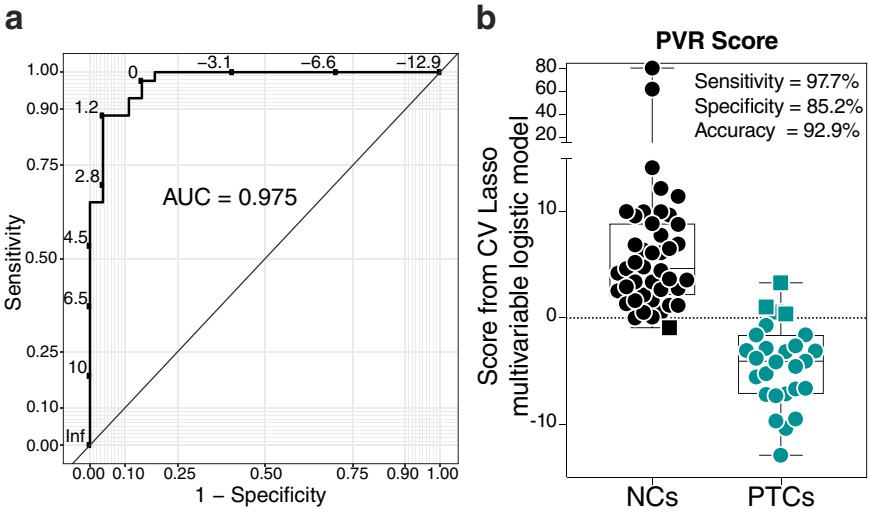

**Fig. 7 A multivariable logistic model using Lasso selected variables predicts the probability of viral remission post-ATI.** The machine-learning algorithm, Lasso regularization, selected seven markers from the ten markers in Fig. 6. Analysis using this model demonstrates that when these seven markers are combined, their predictive ability is better than the predictive ability of any marker individually (Supplementary Table 9). **a** Receiver operator characteristic (ROC) curve showing the area under the curve (AUC) is 97.5% from the multivariable logistic regression model with seven variables. $n = 70$ biologically independent samples. **b** Coefficients from the multivariable logistic model were used to estimate risk score for each individual and then tested for the ability of these scores to accurately classify post-treatment controllers (PTCs) and non-controllers (NCs) at an optimal cut-point. The model correctly classified 97.7 of NCs (sensitivity), 85.2% of PTCs (specificity) with an overall accuracy of 92.9%. Squares represent individuals the model failed to identify correctly. The center of the box showing median with the whiskers going from each quartile (25th and 75th percentiles) to the minimum and maximum, respectively. $n = 70$ biologically independent samples. PVR probability of viral rebound score. Source data are provided as a Source Data file.

using Cox models with a C-index of 74% and probability of viral rebound using a logistic model with an AUC of 97.5%. The utility of these multivariable models to be used in HIV cure-directed clinical trials warrants further investigation. Upon further validation, and possibly combining with other markers, these models could have a profound impact on the HIV cure field by mitigating the risk of ATI during HIV cure-focused clinical trials and provide means for selecting only the most promising therapies and most likely individuals to achieve viral remission to be tested by ATIs.

Beyond their utility as biomarkers, these metabolic and glycomic signatures of viral rebound represent an opportunity to better understand the host milieu preceding a viral rebound. The likelihood of viral rebound and viral remission after ART cessation is likely a function of both the size of the inducible replication-competent HIV reservoir and the host environment that influences inflammatory and immunological responses[2]. The ongoing efforts by many groups to understand the quantitative and qualitative nature of the HIV reservoir are critical to understanding the virological basis of viral rebound[43,44]. However, complementary studies are also needed to understand host determinants of inflammatory and immunological states that may impact post-treatment control of HIV. Our functional analyses on two of these biomarkers (L-glutamic acid and pyruvic acid in Fig. 2) suggest that our signatures have a potential functional significance for post-ART control of HIV. These markers may directly impact latent HIV reactivation or may indirectly condition the host environment with differential levels of inflammation that might impact viral reactivation, cellular processes, and immunological functions during ATI. The potential direct and indirect functional significance of each of the key variables in our models warrants further investigations as they can serve as windows into the mechanisms that contribute to post-ART HIV control.

Our data obtained from two independent cohorts suggest that the bioactive plasma metabolites might not only predict the

duration and probability of viral remission but also actively contribute to it. Our in vivo data showed that the pre-ATI levels of L-glutamic acid predict a delayed viral rebound and a higher probability of viral remission. Indeed, our in vitro validation experiments showed that L-glutamic acid could directly suppress HIV reactivation and suppress LPS and IFNγ-mediated inflammation of myeloid cells. It has been argued that L-glutamic acid, through its conversion to α-ketoglutarate, fuels the TCA cycle/oxidative phosphorylation, which is typically regarded to be an anti-inflammatory metabolic signature[45]. TCA cycle metabolites may regulate immune processes through epigenetic modifications such as DNA methylation[46], which may directly impact proviral reactivation. We also observed significant negative correlations between pre-ATI levels of plasma L-glutamic acid and levels of cell-associated intact and defective HIV DNA in the blood. This is consistent with our in vivo and in vitro data on L-glutamic acid. In contrast to L-glutamic acid, our in vivo data showed that elevated pre-ATI levels of the pro-inflammatory pyruvic acid are associated with an accelerated viral rebound. We also observed a significant positive correlation between pre-ATI levels of plasma pyruvic acid and total HIV DNA. Our in vitro data confirmed these in vivo observations and showed that pyruvate could induce a pro-inflammatory phenotype in myeloid cells upon stimulation. Aerobic glycolysis, where pyruvate is converted into lactate, drives pro-inflammatory M1-macrophage polarization[47], in the context of HIV infection[38]. This is consistent with our in vivo and in vitro data on pyruvate. While no studies have evaluated the impact of plasma metabolic alterations in ATI, one study observed a glycolytic plasma profile in transient HIV elite controllers (TECs) compared to persistent elite controllers (PECs)[48]. Elite controllers are individuals who maintain undetectable levels of viremia in the absence of ART. Moreover, glutamic acid was shown to be elevated in PECs compared to TECs[48], corresponding to our observation that glutamate metabolism was associated with delayed time to HIV rebound. The plasma metabolite signatures we observed are likely a snapshot of the

global and intrinsic cellular metabolic flux that occurs during ATI in HIV-infected individuals.

In addition to L-glutamic acid and pyruvic acid, other intriguing plasma metabolites emerged from the analysis of the Philadelphia cohort. Among the plasma markers associated with delayed viral rebound is ethylmalonic acid. Ethylmalonic acid is central in the metabolism of butyrate, a short-chain fatty acid produced by the gut microbiota and known for its anti-inflammatory effects[49]. Another group of metabolites, consisting of indole-3-pyruvic acid, indole-3-lactic acid, 3-indoxyl sulfate, and 2-oxindole, characterized accelerated rebound and may reflect a biochemical manifestation of dysbiosis of gut bacteria resulting in tryptophan catabolism[50]. Indeed, the tryptophan metabolic pathway was highlighted as one of the main metabolic pathways associated with an accelerated viral rebound. Although it was not mechanistically interrogated, a positive association between plasma indoleamine 2,3-dioxygenase (IDO) activity (an immunoregulatory enzyme that metabolizes tryptophan) and total HIV DNA in peripheral blood has been established[51]. Impaired intestinal barrier integrity is a classical feature of HIV infection, characterized by dysbiosis and increased microbial by-products that drive systemic and mucosal inflammation[52]. Microbes with the capacity to catabolize tryptophan have been linked to adverse HIV disease progression[53], at least in part due to induction of IDO1 that interferes with Th17/Treg balance in the periphery and gut[54]. Our data highlight previously unrecognized interactions between the gut microbiome, its metabolic activity, and HIV persistence. Understanding these potential multi-nodal complex relationships during ART and post-ATI warrants further investigation.

Similar to metabolites, glycans on glycoproteins are bioactive molecules and can play significant roles in mediating immunological functions. For example, antibody glycans can alter an antibody's Fc-mediated innate immune functions, including ADCC and several pro- and anti-inflammatory activities[14–16]. Among glycans on antibodies, the presence of core fucose results in a weaker binding to Fcγ receptor IIIA and reduces ADCC[55]. The same occurs with terminal sialic acid, which reduces ADCC[56]. On the other hand, terminal galactose induces ADCC[57]. In three independent geographically distinct cohorts, two studied in our previous pilot study[22] and the ACTG cohort studied in the current study, we observed a significant association between pre-ATI levels of the digalactosylated non-fucosylated non-sialylated glycan, G2, and delayed viral rebound. G2 is the only IgG glycan trait that is terminally galactosylated, non-fucosylated, and non-sialylated (Supplementary Fig. 6) which is compatible with higher ADCC activity. Similar to our pilot study[22], we observed a link between plasma levels of N-acetyl-glucosamine (GlcNAc) and delayed viral rebound. GlcNAc has been reported to have an anti-inflammatory impact during several inflammatory diseases by modulating NFκB activity[58]. Investigating the potential direct impact of these glycans on innate immune functions and inflammation, and how this affects HIV control during ART, warrants further investigations.

Glycoproteins can also be shed from cells in different organs; therefore, their characteristics can reflect these cells' functions. Glycans on the cell surface are involved in signaling cascades controlling several cellular processes[59,60]. It is not clear how the higher pre-ATI levels of plasma fucose, which associate with both delayed viral rebound (in our pilot study[22] as well as the current study) and higher likelihood for PTC status post-ATI, can directly impact viral control during ATI. Nor is it clear how the higher pre-ATI levels of plasma GalNAc-containing glycans (T/Tn antigens), which associate with both an accelerated viral rebound and a lower likelihood for PTC status, can directly impact viral control during ATI. However, these higher levels might reflect differential levels of these glycans on cells in different organs. For example, T-antigens (tumor-associated antigen) and Tn antigen are O-glycans that are truncated and have incomplete glycosylation, commonly present in cancerous cells, and have been used as tumor markers[61,62]. These GalNAc-containing glycans expressed on some normal immune cells (such as T cells) are ligands of the macrophage galactose type lectin (MGL) that is expressed on activated antigen-presenting cells (APCs). MGL interacts with GalNAc-containing glycans on T cells to induce T cell dysfunction[63]. Our data show that higher levels of these antigens in plasma are associated with an accelerated rebound and a lower likelihood of viral remission and raise the question of whether these glycan levels reflect an immunosuppressive environment in NCs and those who rebound fast. Future studies are needed to examine the direct impact of these glycans on HIV control and/or the potential meaning of their levels as reflections of cellular functions in different tissues during ATI in HIV+ individuals.

We ensured the inclusion in our multivariate models of only metabolic and glycomic markers whose significance was not dependent on several demographic and clinical confounders such as age, sex, ethnicity, ART initiation during early versus chronic stages of HIV infection, duration of ART, and pre-ATI CD4 count, as all of these markers can influence HIV reservoir size and/or our metabolic/glycomic signatures. However, other potential confounders could impact our results, including ART regimen, diet, co-morbidities, co-infections, and other medications. Investigating these other confounders as well as investigating geographically distinct and pediatric cohorts should be the subject of future studies. We examined the links between our glycomic and metabolic signatures and levels of cell-associated HIV DNA and RNA in the blood. It will also be important, in future studies, to examine the potential links between these plasma markers and both HIV reservoirs and host immunological and inflammatory responses in blood and tissues (the main site for HIV persistence). Despite these shortcomings, our study identified a set of a non-invasive, previously unrecognized, class of plasma molecules (glycans and metabolites) that could be used as biomarkers of HIV remission. These signatures of viral rebound were obtained using two independent cohorts of ATI and after applying stringent criteria to avoid the potential impact of several confounders. Our exploratory machine-learning algorithms also identified a combination of these markers that can enhance their predictive value. Our signatures, upon further validation, have the potential to fill a major gap in the HIV cure field through their usage as biomarkers of viral rebound during HIV cure-focused clinical trials. In addition, these results may open mechanistic avenues to better understand the fundamental biological processes, including carbohydrate metabolism, that may regulate HIV control during ART and post-ATI.

## Methods

**Study cohorts and ethics**. Analyses were performed from banked plasma samples of two different cohorts that underwent analytical treatment interruption (ATI): (1) Philadelphia Cohort and (2) ACTG cohort. All analyses were performed on samples collected shortly before ATI in both cohorts.

In the Philadelphia cohort[22,23], 24 HIV-infected individuals on suppressive ART underwent an open-ended ATI without concurrent immunomodulatory agents[22,23]. Approval of this study protocol was obtained from the institutional review board (IRB) of the Wistar Institute (IRB# 2303192-2). All participants provided written consents to use their samples and indirect identifiers (e.g., age, sex, ethnicity, and clinical data) for HIV-related research. Time-to-viral-rebound was identified in this cohort as time to VL of 50 copies/ml. Demographic and clinical data on this cohort is in Supplementary Table 1.

The ACTG cohort combined 74 HIV-infected ART-suppressed participants who underwent ATI from six ACTG ATI studies (ACTG 371[24], A5024[25], A5068[26], A5170[27], A5187[28], and A5197[29]). ACTG 371 was a single-arm prospective, stratified trial of four-drug intentionally interrupted ART in acute or recent HIV infection. A total of 121 patients were enrolled in this study in 15 ACTG sites. All patients signed an informed consent approved by each institution IRB and the

National Institute of Allergy and Infectious Diseases (NIAID)[24]. ACTG A5024 was a partially blinded, randomized phase II trial conducted to test four interventional arms involving continued ART plus ALVAC vCP1452 (or placebo) with or without interleukin (IL)-2. Treatment interruption was then conducted to assess HIV control. A total of 81 patients were enrolled in this study in 19 ACTG sites. The study was approved by on-site IRBs[25]. ACTG A5068 was a prospective, randomized, partially double-blinded study to investigate the effects of immunization with an exogenous HIV vaccine and pulse exposure to the patient's unique viral epitopes on the dynamics of viral rebound after treatment interruption. A total of 97 patients were enrolled in this study in 15 ACTG sites. The study protocol was approved by local IRBs[26]. ACTG 5170 was a multicenter, observational, prospective study of HIV-infected patients receiving ART who had CD4 counts >350 cells/mm[3] and underwent treatment interruptions without interventions. A total of 167 patients were enrolled in this study in 26 ACTG sites. The study was approved by IRBs at each site[27]. ACTG A5187 was a phase I/II, randomized, placebo-controlled, double-blinded trial to evaluate the safety and immunogenicity of an HIV-1 DNA vaccine (VRC-HVDNA 009-00-VP) in patients treated with ART during acute/early HIV-1 infection. A total of 20 patients were enrolled in this study in five ACTG sites. The study was approved by IRBs at each site[28]. ACTG A5197 was a double-blinded study where participants were randomized 2:1 to receive a replication-defective Ad5 vaccine containing HIV-1 gag insert or a placebo. A total of 114 patients were enrolled in this study in 26 ACTG sites. The study was approved by IRBs at each site[29].

These six ATI studies from ACTG included 600 participants and identified 27 PTCs among their participants. Our ACTG cohort included all 27 PTCs and 47 matched non-controllers (NCs) from the same studies. The definition of post-treatment control was: remaining off ART for ≥24 weeks post-ATI with VL ≤ 400 copies for at least 2/3 of time points; no ART in the plasma; and no evidence of spontaneous control pre-ART. The 47 NCs rebounded before meeting PTC criteria[30,31], These two groups were matched for sex, age, % treated at the early stage of HIV infection, ART duration, pre-ATI CD4 count, and ethnicity, as shown in Table 1. Full demographic and clinical data on this cohort is in Supplementary Table 2. All participants provided written consents to use their samples and indirect identifiers (e.g., age, sex, ethnicity, and clinical data) for HIV-related research.

**Plasma untargeted metabolomics analysis**. Metabolomics analysis was performed as described previously[64]. Briefly, polar metabolites were extracted from 50 μl plasma samples with 500μl ice-cold 80% methanol, and deproteinated supernatants were stored at −80 °C prior to analysis. A quality control (QC) sample was generated by pooling equal volumes of all samples after extraction. LC-MS analysis was performed on a Thermo Scientific Q-Exactive HF-X mass spectrometer with HESI II probe and Vanquish Horizon UHPLC system. Hydrophilic interaction liquid chromatography (HILIC) was performed at a flow rate of 0.2 ml/min on a ZIC-pHILIC column (150 × 2.1 mm, 5 μm particle size, EMD Millipore) with a ZIC-pHILIC guard column (20 × 2.1 mm, EMD Millipore) at 45 °C. Solvent A was 20 mM ammonium carbonate, 0.1% ammonium hydroxide, pH 9.2, and solvent B was acetonitrile. The gradient was 85% B for 2 min, 85% B to 20% B over 15 min, 20% B to 85% B over 0.1 min, and 85% B for 8.9 min. The autosampler was held at 4 °C. For each analysis, 4 μl of the sample was injected. The following parameters were used for the MS analysis: sheath gas flow rate, 40; auxiliary gas flow rate, 10; sweep gas flow rate, 2; auxiliary gas heater temperature, 350 °C; spray voltage, 3.5 kV for positive mode and 3.2 kV for negative mode; capillary temperature, 325 °C; and funnel RF level, 40. All samples were analyzed by full MS with polarity switching. The QC sample was analyzed at the start of the sample sequence and after every 8–14 samples. The QC sample was also analyzed by data-dependent MS/MS with separate runs for positive and negative ion modes. Full MS scans were acquired at 120,000 resolution with an automatic gain control (AGC) target of 1e6, maximum injection time (IT) of 100 ms, and scan range of 65–975 $m/z$. Data-dependent MS/MS scans were acquired for the top 10 highest intensity ions at 15,000 resolution with an AGC target of 5e4, maximum IT of 50 ms, isolation width of 1.0 $m/z$, and stepped normalized collision energy (NCE) of 20, 40, 60.

Data analysis was performed using Compound Discoverer 3.1 (ThermoFisher Scientific) with separate analyses for positive and negative polarities. Retention time alignment used the adaptative curve model with 0.3 min maximum shift, 5 ppm mass tolerance, and 3 S/N threshold. Peak detection required less than 5 ppm mass error for extracted ion chromatograms with a 50,000 minimum peak intensity. [M + H] + 1 and [M-H]-1 were set as base ions with consideration for other adducts. Peaks were required to have a width at half height less than 0.5 min and a minimum of 5 scans. Components that had only a monoisotopic peak and no further isotopes were not considered. The maximum element count for isotope pattern modeling was $C_{90}H_{190}N_{10}Na_2O_{15}P_3S_5$. Compounds were grouped across samples with 5 ppm mass error and 0.3 min retention time shift. Peaks not detected initially in a given sample were determined using the fill gaps algorithm with 5 ppm mass error and 1.5 S/N threshold with real peak detection. The gap function uses a priority system to determine missing values: (1) matching detected ions based on expected m/z and retention time regardless of adduct assignment, (2) re-detecting peaks at lower thresholds, (3) simulating peaks based on expected $m/z$, and (4) imputing spectrum noise based on detection limit values. Compound quantifications were corrected for instrument drift by QC areas using the cubic

spline regression model. Each compound was required to be detected in at least 40% of QC runs with a Relative Standard Deviation (RSD) less than 50%. RSD values of the significant metabolites in both the Philadelphia and ACTG Cohorts are shown in Supplementary Table 11. Metabolites were identified by accurate mass (5 ppm mass error) and retention time (0.5 min shift) using a database generated from pure standards or by accurate mass and MS2 spectra using the mzCloud spectral database (mzCloud.org), specifically the 'Endogenous Metabolites' and 'Steroids/Vitamins/Hormones' compound classes and selecting the best matches with HighChem HighRes identity search match factors of 50 or greater. Results were manually processed to remove entries with apparent peak mis-integrations and correct commonly misannotated metabolites. Positive and negative data sets of identified compounds were merged, and the preferred polarity was selected for compounds identified in both polarities. Compound quantifications were normalized per volume plasma injected, which was equivalent for all samples. Values from the ACTG study were further normalized to the summed area of identified metabolites in each sample. For compounds identified multiple times at different retention times, a single entry was selected with priority given to standards database matches followed by greater mzCloud match factors and peak areas.

**In vitro examination of the impact of ʟ-glutamic acid on latent HIV reactivation**. J-Lat cells were used as a model of HIV latency. J-Lat cells harbor latent, transcriptionally competent HIV provirus that encodes green fluorescent protein (GFP) as an indicator of viral reactivation[36]. Levels of latent HIV transcription after stimulation can be measured using flow cytometry. ʟ-glutamic acid was purchased from Sigma (catalog# 49449-100G) and was dissolved in cell-culture compatible HCl solution (Sigma catalog# H9892-100ML). J-Lat 5A8 clone was kindly provided by Dr. Warner Greene (The Gladstone Institute of Virology and Immunology). J-Lat clone 10.6 (catalog number 9849) was provided by the NIH AIDS Reagent Program (Germantown, MD). Cells from different clones of J-Lat (5A8 and 10.6) were cultured at $1 \times 10^6$ cells/ml in cultured in R10 media (complete RPMI 1640 medium supplemented with 10% fetal bovine serum (FBS, penicillin (50 U/ml), and streptomycin (50 mg/ml) and were stimulated with PMA/iono-mycin (16 nM/500 nM- Sigma catalog# P8139/ catalog# I0634-1MG, respectively) or ImmunoCult Human CD3/CD28 T Cell Activator (Stem cell catalog# 10971), or TNFα (10 ng/ml; Stem Cell catalog# 78068.1) in the presence of HCl solution as a control. J-Lat cells were also treated with ʟ-glutamic acid (5 mM) in the presence or absence of the above stimulators. After 24 h, cells were stained with live/dead marker (Thermo catalog# L34966), and GFP Mean Fluorescence intensity (MFI) was measured by LSR II flow cytometer and FACSDiva software. To test the impact of glutamine in the RPMI media on our results, J-Lat 5A8 cells were cultured in glutamine-free media (Gibco, catalog # 21870076, supplemented with 10% FBS, penicillin (50 U/ml), and streptomycin (50 mg/ml)) and treated with ʟ-glutamic acid (4, 5, or 6 mM) in the presence or absence of PMA/I (HCL solution was used as a negative control). After 24 h, cells were stained with live/dead marker, and GFP MFI was measured by LSR II flow cytometer and FACSDiva software. The gating strategy for the J-Lat experiments is shown in Supplementary Fig. 8 using FlowJo software (version 10.7.01).

**In vitro examination of the impact of ʟ-glutamic acid and pyruvate on myeloid inflammation**. THP-1 cell line (catalog number 9942) was provided by the NIH AIDS Reagent Program (Germantown, MD). THP1 cells were plated in 24-well plates at a density of $7 \times 10^5$ cells per well. To differentiate them into macrophage-like, 100 nM of PMA (Sigma catalog# P8139) was added and incubated for 72 h. After incubation, media was aspirated, and each well was gently washed twice with R10 media. Cells were then rested for 24 h on R10 media without PMA. After 24 h, cells were washed again with serum-free (no FBS) RPMI 1640 media and kept in this media for the rest of the experiment. Macrophage-like THP1 cells were pre-incubated with ʟ-glutamic acid (5 mM) or Sodium Pyruvate solution (2 mM, Sigma catalog# S8636-100ml) for 2 h before stimulating with Escherichia coli serotype O127:B8 LPS (50 ng/ml; Sigma catalog# L3129-10MG) and IFNγ (10 ng/ml; R&D Systems catalog# 285-IF-100, respectively). After 5 h of incubation with LPS/IFNγ, culture supernatants were collected for cytokine quantitation. Supernatant levels of IL-10, IL-12p70, IL-13, IL-1β, IL-2, IL-4, IL-6, and IL-8 were determined using U-PLEX Proinflam Combo 1 (Meso Scale Diagnostic # K15049k-1) according to manufacture. Levels of TNFα were quantified using DuoSet ELISA kits (R&D Systems; Catalog# DY210-05).

**IgG isolation**. Bulk IgG was purified from 50 μl plasma using Pierce™ Protein G Spin Plate (Thermo Fisher catalog# 45204). IgG purity was confirmed by SDS gel.

**N-glycan analysis using capillary electrophoresis**. For both plasma and bulk IgG, N-glycans were released using peptide-N-glycosidase F (PNGase F) and labeled with 8-aminopyrene-1,3,6-trisulfonic acid (APTS) using the GlycanAssure APTS Kit (Thermo Fisher cat. A33952), following the manufacturer's protocol. Labeled N-glycans were analyzed using the 3500 Genetic Analyzer capillary electrophoresis system. IgG N-glycan samples were separated into 22 peaks and total plasma N-glycans into 24 peaks. Relative abundance of N-glycan structures was quantified by calculating the area under the curve of each glycan structure divided

by the total glycans using the Applied Biosystems GlycanAssure Data Analysis Software Version 2.0.

**Glycan analysis using lectin array**. To profile the plasma total glycome, we used the lectin microarray as it enables analysis of multiple glycan structures; it employs a panel of 45 immobilized lectins with known glycan-binding specificity. Plasma proteins were labeled with Cy3 and hybridized to the lectin microarray. The resulting chips were scanned for fluorescence intensity on each lectin-coated spot using an evanescent-field fluorescence scanner GlycoStation Reader (Glyco-Technica Ltd.), and data were normalized using the global normalization method.

**Quantification of HIV DNA and CA-RNA**. Cell-associated (CA)-RNA and DNA were isolated from cryopreserved peripheral blood mononuclear cells (PBMCs) using the AllPrep DNA/RNA Mini Kit (Qiagen). Unspliced CA-RNA and total HIV DNA levels were quantified using a real-time PCR approach with primers/ probes targeting conserved regions of HIV LTR/gag as previously described (Supplementary Table 12)[6,65]. The CA-RNA assay measures levels of unspliced transcripts, which are late RNA products necessary for the creation of HIV structural proteins and remains one of the most commonly used assay in HIV curative studies[66–68]. Cell numbers were quantified by the real-time PCR measurement of CCR5 copy numbers. Cellular integrity for RNA analysis was assessed by the measurement of total extracted RNA and evaluation of the IPO-8 housekeeping gene[69].

**Near-full-length HIV proviral sequencing**. Single-genome, near-full-length proviral sequences were obtained from a previously published dataset generated on a subset of the ACTG cohort[31]. Briefly, limiting-dilution proviral amplification was performed, and DNA was extracted from PBMCs using the QIAmp DNA Mini Kit (Qiagen). Isolated DNA was amplified using limiting-dilution nested PCR amplification (Supplementary Table 12)[31]. PCR amplicons were sequenced using the Illumina MiSeq platform. A continuous fragment of HIV-1 proviral DNA was assembled, and the sequences were aligned to HXB2 to identify sequence defects (e.g., internal deletions, premature stop codons, out-of-frame mutations, internal inversions, and packaging signal defects). The sequences were also tested for hypermutations using the Los Alamos HIV Sequence Database Hypermut program to identify hypermutated sequences. Proviral sequences that lacked the above-mentioned defects were classified as intact[31].

**Statistical analysis**. For each of the studied biomarkers, data distribution was first examined, and appropriate data transformation was made for further analysis. Data from metabolic analysis were log$_2$-transformed before analysis. Data from the lectin array were log$_2$-transformed in the Cox and logistic regression analyses (Figs. 3, 4 and 7, as well as Supplementary Tables 6, 7, 8, and 10). Original data from the lectin array were used in the analysis for Fig. 6 and Supplementary Table 9. Two-group t-tests or Mann–Whitney tests were used to determine the difference between two groups. Spearman's rank correlation coefficient was used to evaluate correlations. For binary outcome (NCs vs. PTCs) or time-to-viral-rebound, logistic or Cox regression models with or without adjusting for confounders were used to assess the association between a biomarker and outcome, respectively. False discovery rates (FDR) were calculated using Benjamini–Hochberg correction. To explore biomarkers that could be predictors of clinical outcomes, specific sets of biomarkers were identified among those with FDR < 0.1. Variables for the multivariable models were selected from the identified specific sets of biomarkers using the Lasso technique with the cross-validation (CV) selection option by separating data in 5-fold. Due to this exploratory study with a modest sample size, variables selection was determined using 100 independent rounds runs of CV Lasso with minimum tuning parameter lambda. The biomarkers that were selected 80 or more times from 100 runs were used as a final set of predictors in our models. The predictive ability of the final logistic and Cox models was assessed by AUC and C-index. GraphPad Prism 7, Stata 16, and R were used for data analysis.

**Reporting summary**. Further information on research design is available in the Nature Research Reporting Summary linked to this article.

## Data availability

The authors declare that data supporting the findings of this study are available within the paper and its supplementary information files. Source data are provided with this paper. The metabolomics data generated in this study have been deposited in the NIH Common Fund's National Metabolomics Data Repository (NMDR) website, the Metabolomics Workbench, https://www.metabolomicsworkbench.org where it has been assigned Project ID PR001053. The data can be accessed directly via its Project DOI: 10.21228/M8KQ59 (https://doi.org/10.21228/m8kq59). Metabolomics Workbench is supported by NIH grant U2C-DK119886. Source data are provided with this paper.

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

## Acknowledgements

This work is supported by a grant from the Foundation for AIDS Research (amfAR) to M.A.-M and J.Z.L and the NIH R21 AI143385 to M.A.-M. M.A.-M is also supported by NIH grants (R01 DK123733, R01 AG062383, R01NS117458, R21 AI129636, and R21 NS106970), the Penn Center for AIDS Research (P30 AI 045008), and W.W. Smith Charitable Trust. L.J.M is supported by R01AI48398, the NIH-funded BEAT-HIV Martin Delaney Collaboratory to cure HIV-1 infection (1UM1AI126620), Herbert Kean, M.D., Family Professorship, and the Robert I. Jacobs Fund of the Philadelphia Foundation. Metabolomics analysis was performed by the Wistar Proteomics and Metabolomics Shared Resource supported in part by NIH Cancer Center Support Grant CA010815 on a Thermo Q-Exactive HF-X mass spectrometer purchased with NIH grant S10 OD023586. This work was also supported by the National Institutes of Health (NIH) grant UM1 AI068634 to the Statistical and Data Management Center of the AIDS Clinical Trials Group, UM1 AI068636 to AIDS Clinical Trials Group, and a subcontract from UM1 AI106701 to the Harvard Virology Support Laboratory. We would like to thank Rachel E. Locke, Ph.D., for providing comments. We would like to thank all donor participants. We thank the participants, staff, and principal investigators of the ACTG studies A371, A5024, A5068, A5170, A5187, and A5197.

## Author contributions

M.A.-M. conceived and designed the study. L.B.G. carried out the majority of experiments. C.S.P. analyzed and interpreted metabolic data. M.D. ran the lectin array experiments. E.P., R.R.S., B.E., R.J., K.M., J.R.K., P.T., A.L., L.J.M., J.J., and J.Z.L. selected study participants and interpreted clinical data. A.R.G. and H.T. performed metabolic analysis. X.Y. and Q.L. performed statistical analysis for the whole study. L.B.G., C.S.P., and M.A.-M. wrote the manuscript, and all authors edited it.

## Competing interests

The authors declare no competing interests.
