## [Peer Review File · Nature Communications]

REVIEWER COMMENTS

Reviewer #1 (Remarks to the Author):

The authors investigate whether they can use metabolic and glycomic biomarkers to predict post-treatment controllers in HIV antiretroviral therapy. They show convincing evidence that they have identified some metabolic markers linked to inflammatory pathways that are positively or negatively associated with PTC. The paper is well written and I believe it will be of interest to readers of Nature Communications.

I did try and look at the data loaded into Metabolomics Workbench. I had been sent two links for each study. The first link took me to the correct dataset, but there was no access to raw data. There were also no QC samples listed which I feel should be corrected before publication.

The second link, sent later, could not be properly opened. A google search kept on turning up a study on breast cancer with that number. This problem should not prevent acceptance of the paper, but should ideally be corrected before publication.

I have a few minor points:

The ACTG was a combination of 6 different AGTC studies – were the PTCs split relatively evenly across all of the studies. Was study group membership tested as a confounding factor?

Line 127: please give the total number of time points analysed here.

Line 144: an FDR of 20% is used. In line 237, it has now been reduced to 10%. Why?

Line 419: you may wish to explain what elite controls are for a wider audience.

Line 519: Please give all processing details of your metabolomics data, including normalization and missing value substitution in line with the Metabolomics Society recommendations on reporting guidelines.

Line 521: were QC samples pooled before or after extraction. What were the RSDs for the QC samples for the metabolites that were considered clinically important?

Line 572: did you normalize the cell cultures in any way?

Line 610: given the unequal group sizes, were there any restrictions placed on the cross validation groups?

Fig 1 C and D and supp table 3: as I understand your pathway analysis, you have only included compounds that you had already determined were significant. This limits the number of pathways

that can be represented here and gives very little information on which pathways (as opposed to individual metabolites) may be the most significant. The approach may hide pathways which have persistent but small changes across the entire pathway. Have you tried including all detected metabolites and conducting a pathway analysis based on using a lower criteria for significance e.g. significant p value before false discovery correction? Does this still identify these same pathways as the most important?

Fig 2: please use darker colours for your error bars. They are not always visible.

Fig 6: did you explore why individuals were misclassified? And did the degree of classification reflect the individual's viral load, especially for individuals who were close to the PTC classification cut off line?

I confess I don't understand supp table 8. The authors state that they controlled for a range of confounding factors, but they had very few PTCs in comparison to the NCs, and the overall group size is small. Some of the their reported compounds did seem to be influenced by confounding factors – how was this then corrected? Are the stated p values before or after correction? What are the confidence intervals?

Reviewer #2 (Remarks to the Author):

Several HIV therapeutic strategies are designed to reduce the magnitude of the HIV reservoirs and enhance the immune system, aiming to promote antiretroviral therapy-free HIV remission. However, the lack of reliable biomarkers to assess virological control poses as a major challenge to monitor the success of such interventions. Analytical treatment interruption (ATI) has been considered the gold standard approach to evaluate the effectiveness of such strategies. Although ATI studies with close monitoring are well-tolerated by participants, there are several concerns in terms of safety and long-term impact on the individuals and communities. Investigation of reliable biomarkers that could be used to predict HIV rebound dynamics after ATI, could be instrumental to inform and guide clinical decisions during ATI studies, reduce risks and contribute to the development successful HIV cure strategies. Studies to investigate such biomarkers are highly relevant and could significantly advance the field of HIV cure research.

In their study “Non-Invasive Plasma Glycomic and Metabolic Biomarkers of Post-treatment Control of HIV” Giron et al. evaluated two cohorts of people living with HIV who were on suppressive ART and underwent ATI, namely the Philadelphia Cohort and the ACTG cohort, to identify potential pre-ATI biomarkers that could predict viral rebound dynamics post-ATI. First, the authors evaluated stored plasma samples (n=24 HIV+ participants) from the Philadelphia cohort during ATI via untargeted metabolomics and glycomics. Using the same methods, the team subsequently

evaluated stored plasma samples pre-ATI from a total of 74 participants selected from 6 ACTG clinical trials, including 27 participants that met criteria as post-treatment controllers (PTC) and 47 participants matched non-controllers (NC). This larger cohort was used as validation cohort. The study identified a set of pre-ATI plasma glycans and metabolites that could predict time-to-viral-rebound and probability-of-viral-rebound. Using machine-learning models the study identified the smallest set of pre-ATI biomarkers that could predict time-to-viral-rebound with 74-76% capacity and probability-of-viral-rebound with 97.5% capacity in the ACTG cohort.

The study topic is highly significant to the field of HIV and I highlight below several aspects that I considered strengths of this work.

1. The manuscript is well-written and clearly describes the rationale, study design and discussion of the findings. The statistical methods used are standard, and appropriate for this type of study.
2. Untargeted metabolomics as it is an unbiased global analysis of small molecules can be instrumental for discovery of new biomarkers associated with clinical conditions, with potential translational applications. A growing body of research is taking advantage of this methodology. The application of untargeted metabolomics in the context of ATI studies is novel and could add important new information to the field.
3. The use of well characterized cohorts, including a validation cohort is another strong aspect of this study.
4. Assessing the potential effect of selected top candidate metabolites biomarkers via a functional in vitro assay supported the idea that some of these markers could act via a direct effect on HIV reservoirs dynamics.
5. The authors also evaluated PTC participants, as a model for HIV control, compared to well-matched NC, which offered a great opportunity to identify potential biomarkers associated with control of HIV after ART interruption.

Overall, the findings have the potential to contribute to both, HIV cure strategies research as well as providing new insights into potential mechanisms underlying viral control in people living with HIV. A few points for clarification and further consideration are listed:

1. A powerful aspect of untargeted metabolomics is that it enables the detection thousands of known and unknown small molecules from a biological sample. Usually only about 10% can be identified using currently available (but growing) databases. The study reports that 179 and 226 metabolites were identified in plasma samples from Philadelphia and ACTG cohorts, respectively. Was the metabolomics data filtered in any ways prior to the statistical analysis? I am also curious if the authors have considered evaluating the global metabolomics profiles, including the unknown molecules. I would expect some of the unknown molecules to be significantly associated with the tested outcomes.

2. Plasma samples evaluated in this study are reported to be from timepoints pre-ATI for the ACTG cohorts. What was the time interval between pre-ATI sample collection and ATI? Were PBMC samples paired to pre-ATI plasma?
3. Were levels of pre-ATI ca-HIV DNA and RNA associated with time-to-viral and probability of virus rebound in these cohorts? Were there differences between PTC and NC included in this study?
4. Given the heterogeneity of HIV reservoirs, one important limitation was the lack of information on levels of replication-competent or inducible reservoirs pre-ATI. I recognize that the ability of doing such assays is hampered by the lack of freshly collect and/or large number of cells required (as it is the case of QVOA or TILDA assays), in studies where only stored frozen samples are available. Although less ideal, assays such as ddPCR-based IPDA or near-Full-length sequencing, that can measure levels of intact versus non-intact reservoirs to be used as a proxy of intact and likely replication-competent provirus, could have partially addressed this limitation. I wonder if this type of data has been generated as part of the parent studies or, could be generated as part of this study to evaluate its association with plasma markers and viral rebound dynamics.
5. Were the selected PTC and NC participants equally distributed over each of the parent ACTG clinical trials?
6. Why did the authors decide not to include pre-ATI ARV regimen as a potential cofounder in the models, if this data was available?
7. Is gender in the manuscript used in place of biological sex? If so, I would suggest replacing gender for sex in the manuscript/figures/tables.
8. There are a few remaining typos throughout the manuscript, I would suggest a careful revision.
9. The study findings were very interesting and encourage further studies to identify potential combination of markers that could help predict viral rebound dynamics in HIV cure studies with ATI. Overall, the authors did a good job on highlighting that the potential use of these signatures to guide clinical trials requires further validation, with exception of a very few remarks stating these biomarkers “can be used” to access viral remission. I suggest being careful to avoid overstatements. I would suggest changing this to “could be used”.
10. What do the authors think would be the next steps to further validate the molecules identified in this study as biomarkers of viral remission?

Reviewer #3 (Remarks to the Author):

Giron et al's 'Non-Invasive Plasma Glycomic and Metabolic Biomarkers of Post-treatment Control of HIV' nicely written report on multiple cohorts motivates the need for non-invasive biomarkers that can predict HIV remission after ART interruption. They identified a number of glycomic and

metabolic signatures associated at the univariate and leveraging a penalized methodology in combination a signature of sorts that predicts time to viral rebound and probability of viral rebound. However, I have a few questions on the statistical methodology.

Major comments:

- In my reading it was not clear if the complete dataset (n=70, 4 without complete data) was used in training this algorithm from 2 data sets?

If this is the case, then the AUC and c-index values reported is likely an overfit of the model and would not replicate. An alternative approach to analyzing the data would be to take the first cohort - train the model - and then test on the second cohort.

Another approach could include a K-fold cross validated model where then the variance of the model estimates are reported and the prediction measures for each of the hold-out folds are reported. This would ensure a more robust modelling.

- My second comment is really that a C-index of 0.74 is good, but at least in many different settings is not enough for clinical improvement or change. Could the authors clarify why they believe a ~0.75 c-index is clinically meaningful?

We are very grateful to the reviewers for providing us with constructive and consistent feedback and for giving us the opportunity to revise and improve our manuscript entitled “*Non-Invasive Plasma Glycomic and Metabolic Biomarkers of Post-treatment Control of HIV.*”

We have made several modifications throughout the manuscript, which we believe address all reviewers' concerns. We have included detailed, point-by-point responses to each of the reviewers' concerns (below). Our responses are in blue text to facilitate the review process. We have also included a revised version of the manuscript with changes highlighted. We believe that our manuscript is now significantly improved as a result of the reviewers' constructive comments.

Reviewer #1

Reviewer general comment: The authors investigate whether they can use metabolic and glycomic biomarkers to predict post-treatment controllers in HIV antiretroviral therapy. They show convincing evidence that they have identified some metabolic markers linked to inflammatory pathways that are positively or negatively associated with PTC. The paper is well written and I believe it will be of interest to readers of Nature Communications.

Authors response: We appreciate the reviewer's opinion of our findings and that he/she found them of interest. As detailed below, we have now addressed all concerns raised with modifications throughout the manuscript.

Specific comments:

1. I did try and look at the data loaded into Metabolomics Workbench. I had been sent two links for each study. The first link took me to the correct dataset, but there was no access to raw data. There were also no QC samples listed which I feel should be corrected before publication. The second link, sent later, could not be properly opened. A google search kept on turning up a study on breast cancer with that number. This problem should not prevent acceptance of the paper, but should ideally be corrected before publication.

Authors response: We have now updated the entry into the *Metabolomics Workbench* and added QC samples. This update is now highlighted in the revised manuscript as:

“Metabolomics data are available at the NIH Common Fund's National Metabolomics Data Repository (NMDR) website, the Metabolomics Workbench, <https://www.metabolomicsworkbench.org> where it has been assigned Project ID PR001053. The data can be accessed directly via its Project DOI: 10.21228/M8KQ59. Metabolomics Workbench is supported by NIH grant U2C-DK119886.”

These data can be accessed through these links:

Philadelphia Cohort:

https://www.metabolomicsworkbench.org/data/MWTABMetadata4.php?Mode=Study&DataMode=AllData&StudyType=MS&F=MohamedMohsen_20210311_120513_mwtab_analysis_1.txt

ACTG Cohort:

https://www.metabolomicsworkbench.org/data/MWTABMetadata4.php?Mode=Study&DataMode=AllData&StudyType=MS&F=MohamedMohsen_20210311_123623_mwtab_analysis_1.txt

Username: MohamedMohsen
Password: PTCHIV2020!

2. The ACTG was a combination of 6 different AGTC studies – were the PTCs split relatively evenly across all of the studies. Was study group membership tested as a confounding factor?

Authors response: **Table 1** in the document, shows the distribution of PTCs and NCs among the six original ACTG studies. We have now added a new **supplementary Table 2** with the demographic and clinical characteristics of each of the 74 participants of the ACTG cohort to make it easier for the reader to explore these important questions. We have also added the study source as a confounder in our analyses (**Supplementary Tables 6 and 9**), as suggested by the reviewer.

Table 1: Distribution of PTC among the ACTG studies

Study	Total (n)	NC (n, (%))	PTC (n, (%))
A371	24	15 (62.5)	9 (37.5)
A5068	21	13 (61.9)	8 (38.1)
A5170	13	8 (61.5)	5 (38.5)
A5197	8	6 (75)	2 (25)
A5187	5	4 (80)	1 (20)
A5024	3	1 (33.3)	2 (66.7)

3. Line 127: please give the total number of time points analyzed here.

Authors response: We thank the reviewer, and we have now added this information to Line # 91-93 of the revised manuscript (the clean version) and lines # 98-99 of the marked version.

4. Line 144: an FDR of 20% is used. In line 237, it has now been reduced to 10%. Why?

Authors response: We elected to use a more stringent statistical plan to analyze the data from the ACTG cohort compared to the plan used to analyze the data from the Philadelphia cohort. There are two reasons to adopt the more stringent statistical plan: 1) the bigger sample size of the ACTG cohort compared to the Philadelphia cohort; and 2) to ensure high stringency in this later stage of the study. Thus, in the more stringent analysis, we only considered markers with a hazard ratio (HR) ≥ 2 or ≤ 0.5 . We also included only glycomic and metabolic markers with either FDR < 10% or markers that had emerged from the Philadelphia cohort. Lastly, we only included markers that remained significant after adjusting for key demographic and clinical confounders. These three additional criteria were not possible in analyzing the Philadelphia Cohort data due to the smaller sample size.

5. Line 419: you may wish to explain what elite controls are for a wider audience.

Authors response: We thank the reviewer for this suggestion, and we have now added the definition of elite controllers to the text.

6. Line 519: Please give all processing details of your metabolomics data, including normalization and missing value substitution in line with the Metabolomics Society recommendations on reporting guidelines.

Authors response: We thank the reviewer for this suggestion and have thoroughly updated the metabolomics method section with processing details, as requested.

7. Line 521: were QC samples pooled before or after extraction. What were the RSDs for the QC samples for the metabolites that were considered clinically important?

Authors response: The QC samples were pooled after extraction and run periodically during each sample set (after every 8-14 samples depending on sample set); these details are now also added to the methods section. We have also added new **Supplementary Table 11** with RSDs for the clinically important metabolites in both cohorts.

8. Line 572: did you normalize the cell cultures in any way?

Authors response: We ensured the usage of the same number of cells and the same volume of cultures. We have also performed the experiments testing the *in vitro* effects of L-glutamic acid using full RPMI media (with glutamine), as well as using media without glutamine (to examine the effects of glutamine in regular RPMI media on our findings). The new data from experiments using a media without glutamine are now in **Supplementary Figure 3**. We have also added the gating strategy of the J-Lat experiments in **Supplementary Figure 8**.

9. Line 610: given the unequal group sizes, were there any restrictions placed on the cross validation groups?

Authors response: We did not set any restrictions on cross-validations (CV). This decision was based on several reasons: 1) Data from the ACTG samples (27 PTCs and 43 NCs) were used for variable selection with the CV Lasso technique. While the group sizes were unequal, it was not a typical situation of data imbalance (for example, having many observations for one group and much fewer observations in the other group). So, we did not think there was a need to place any restrictions in this analysis. 2) We used 5-fold (80% training and 20% testing) cross-validation analyses in which each training set had 56 patients. In the most extreme case, there were at least 13 PTCs (when all 43 NCs were selected in the training set), which is still an acceptable number of patients from both groups to perform logistic regression analyses. 3) To avoid unreliable results from one-time analysis with a modest sample size, we performed 100 independent rounds of CV Lasso with minimum tuning parameter lambda. The biomarkers that were selected ≥ 80 times from the 100 runs were used as a final set of predictors in our models.

10. Fig 1 C and D and supp table 3: as I understand your pathway analysis, you have only included compounds that you had already determined were significant. This limits the number of pathways that can be represented here and gives very little information on which pathways (as opposed to individual metabolites) may be the most significant. The approach may hide pathways which have persistent but small changes across the entire pathway. Have you tried including all detected metabolites and conducting a pathway analysis based on using a lower criteria for significance e.g. significant p value before false discovery correction? Does this still identify these same pathways as the most important?

Authors response: We thank the reviewer for this suggestion. As suggested, we repeated this analysis using all metabolites with P -values < 0.05 . Below figure is a comparison of the analysis included in the study (limited to metabolites with $FDR < 20\%$) and the new analysis (using all metabolites with $P < 0.05$). The two analyses result in a similar set of pathways. The glutamate metabolism remains one of the most significant pathways associated with a delayed rebound, while the pyruvic acid pathway remains one of the most significant pathways associated with a faster rebound.

11. Fig 2: please use darker colours for your error bars. They are not always visible.

Authors response: We thank the reviewer for this suggestion, and we have now made the necessary changes to Figure 2 (as well as to Supplementary Figure 4).

12. Fig 6: did you explore why individuals were misclassified? And did the degree of classification reflect the individual's viral load, especially for individuals who were close to the PTC classification cut off line?

Authors response: We thank the reviewer for these interesting questions. We tried to look at all available parameters to explain this misclassification, and we could not identify any clear patterns. We have now added a new **supplementary Table 2** with all available demographic and clinical characteristics of each participant (and highlighted these misclassified individuals) to make it easier for the reader to explore these important questions. Perhaps other unknown factors (such as diet, medications ...etc.) could be underlying this misclassification or simply these misclassifications reflect some limitations of the model.

13. I confess I don't understand supp table 8. The authors state that they controlled for a range of confounding factors, but they had very few PTCs in comparison to the NCs, and the overall group size is small. Some of their reported compounds did seem to be influenced by confounding factors – how was this then corrected? Are the stated *p* values before or after correction? What are the confidence intervals?

Authors response: We apologize for any confusion. Due to the small sample size of the PTC group, we adjusted for each of these confounders separately. **Supplementary Table 8 (now Supplementary Table 9)** lists the glycomic and metabolic variables (markers) whose *P*-value < 0.05 after adjusting for each of the given confounders (a total of 7 linear regression analyses were performed for each marker). In this table, the *P*-value reported under a given confounder is not for testing that confounder. It is the *P*-value for testing the differences of the average of the studied marker between groups using a linear regression model adjusting for that given confounder. Thus, the stated *P*-value for each marker is after correction for the given confounder.

To avoid confusion, along with the *P*-values, we have now added the estimated difference of the average levels of the studied marker between groups with 95% confidence interval in Supplementary Table 8 (now **Supplementary Table 9**). In addition, Supplementary Table 5 (now **Supplementary Table 6**) has now also been updated with adding the 95% confidence intervals for hazard ratios.

Reviewer #2

Reviewer general comment: The study topic is highly significant to the field of HIV and I highlight below several aspects that I considered strengths of this work.

1. The manuscript is well-written and clearly describes the rationale, study design and discussion of the findings. The statistical methods used are standard, and appropriate for this type of study.
2. Untargeted metabolomics as it is an unbiased global analysis of small molecules can be instrumental for discovery of new biomarkers associated with clinical conditions, with potential translational applications. A growing body of research is taking advantage of this methodology. The application of untargeted metabolomics in the context of ATI studies is novel and could add important new information to the field.
3. The use of well characterized cohorts, including a validation cohort is another strong aspect of this study.
4. Assessing the potential effect of selected top candidate metabolites biomarkers via a functional in vitro assay supported the idea that some of these markers could act via a direct effect on HIV reservoirs dynamics. The authors also evaluated PTC participants, as a model for HIV control, compared to well-matched NC, which offered a great opportunity to identify potential biomarkers associated with control of HIV after ART interruption. Overall, the findings have the potential to contribute to both, HIV cure strategies research as well as providing new insights into potential mechanisms underlying viral control in people living with HIV.

Authors response: We appreciate that the reviewer found our study significant and has the potential to have a high impact on the HIV cure field. We have addressed all concerns raised as described below.

Specific Comments:

1. A powerful aspect of untargeted metabolomics is that it enables the detection thousands of known and unknown small molecules from a biological sample. Usually only about 10% can be identified using currently available (but growing) databases. The study reports that 179 and 226 metabolites were identified in plasma samples from Philadelphia and ACTG cohorts, respectively. Was the metabolomics data filtered in any ways prior to the statistical analysis? I am also curious if the authors have considered evaluating the global metabolomics profiles, including the unknown molecules. I would expect some of the unknown molecules to be significantly associated with the tested outcomes.

Authors response: We thank the reviewer for this comment. Detected features were filtered prior to the statistical analysis. Only metabolites identified by mass and retention time from standards or spectral matches to the mzCloud database were considered. Identified metabolites were further filtered by peak detection quality, and a single polarity was selected for metabolites identified in both polarities. We have not considered evaluating the unknown molecules or other molecules with lower confidence identifications at this time, though it is an intriguing possibility that some of these molecules may associate with tested outcomes. We will continue exploring unknown molecules as annotation databases continue to be updated. However, we elected not to consider these metabolites, in this manuscript, for two reasons: 1) only known metabolites can be advanced, in future work, as potential biomarkers (by measuring them individually or using a multiplex format); 2) one of the main goals of our manuscript is to provide some potential insights (to be tested in the future) on the host pathways that may contribute to post-ART control of HIV. Only known metabolites can provide these insights into the mechanisms of action that underlie our observations.

2. Plasma samples evaluated in this study are reported to be from timepoints pre-ATI for the ACTG cohorts. What was the time interval between pre-ATI sample collection and ATI? Were PBMC samples paired to pre-ATI plasma?

Authors response: We thank the reviewer for this important question, and we have now added a new **supplementary Table 2** with the demographic and clinical characteristics of each participant. In this table, we included the time interval between sampling and ATI. In most cases, sampling was done during the same week of ATI. Yes, PBMCs were paired to the plasma.

3. Were levels of pre-ATI ca-HIV DNA and RNA associated with time-to-viral and probability of virus rebound in these cohorts? Were there differences between PTC and NC included in this study?

Authors response: We thank the reviewer for this important question. We have now added **Supplementary Figure 7** showing that CA-HIV DNA and RNA levels are different between PTCs and NCs and can predict time-to-viral-rebound (as shown in previous studies).

4. Given the heterogeneity of HIV reservoirs, one important limitation was the lack of information on levels of replication-competent or inducible reservoirs pre-ATI. I recognize that the ability of doing such assays is hampered by the lack of freshly collect and/or large number of cells required (as it is the case of QVOA or TILDA assays), in studies where only stored frozen samples are available. Although less ideal, assays such as ddPCR-based IPDA or near-Full-length sequencing, that can measure levels of intact versus non-intact reservoirs to be used as a proxy of intact and likely replication-competent provirus, could have partially addressed this limitation. I wonder if this type of data has been generated as part of the parent studies or, could be generated as part of this study to evaluate its association with plasma markers and viral rebound dynamics.

Authors response: We thank the reviewer for this suggestion. We have now included near-full length sequencing data on intact, defective, and hyper-mutated HIV DNA from 19 individuals (10 PTC and 9 NC; all samples with this kind of data from our ACTG cohort) from the ACTG cohort (**Supplementary Figure 7** and **Figure 5**). The difference in these measurements between PTCs and NCs was recently published (Sharaf R

et al., J Clin Invest. 2018, PMID: 30024859). We have now added the ability of these measurements to predict time-viral-rebound (**Supplementary Figure 7**) and their associations with our plasma biomarkers (**Figure 5**). Intriguingly, one of the lead metabolite markers (L-glutamic acid), which predicted delayed viral rebound in both the Philadelphia and ACTG cohorts and whose pre-ATI levels were higher in PTCs compared to NCs, exhibited negative correlations with levels of intact and defective HIV DNA. In general, markers associated with delayed rebound were associated with a smaller reservoir, and markers associated with faster rebound were associated with a larger reservoir (**Figure 5A**).

In the future, we will need to further evaluate the relationship between our markers and the intact proviral reservoir in a larger number of individuals; however, as the reviewer noted, near-full length sequencing is a very laborious and time-consuming process that requires a large number of cells (something that is not always available from individuals with this rare phenotype of PTC). We are in the process of validating and improving upon the IPDA. However, this assay is still relatively new, and there have been potential deficiencies that have recently been described (Kinloch NN et al., *Nat Commun*. PMID: 33420062) and which we are trying to overcome before using this assay on these precious samples.

We again thank the reviewer for raising this point as we believe the addition of near-full length sequencing data from these 19 individuals added more depth and sophistication to our manuscript.

5. Were the selected PTC and NC participants equally distributed over each of the parent ACTG clinical trials?

Authors response: **Table 1 (above)** shows the distribution of PTCs and NCs among the six original ACTG studies. We have now added a new **supplementary Table 2** with demographic and clinical characteristics of each participant to make it easier for the reader to explore these important questions. In addition, we also added the study source as one of the confounders we adjust for in Supplementary Table 6 and 9.

6. Why did the authors decide not to include pre-ATI ARV regimen as a potential cofounder in the models, if this data was available?

Authors response: ART regimens were very diverse (new **supplementary Table 2**) to be included as a potential confounder. However, we agree with the reviewer that ART regimen is a potentially very important confounder. We have added this to the limitations section of the study. Hopefully, in the future, there will be ART-regimen-controlled ATI studies (such as the ongoing ACTG A5345/A5347s) that would allow answering this important question.

7. Is gender in the manuscript used in place of biological sex? If so, I would suggest replacing gender for sex in the manuscript/figures/tables.

Authors response: We thank the reviewer for this comment. It was biological sex; hence, we replaced gender with sex in all the manuscript/figures/tables, as suggested by the reviewer.

8. There are a few remaining typos throughout the manuscript, I would suggest a careful revision.

Authors response: We apologize for these typos, and we went through the manuscript, figures, and tables and we tried our best to carefully fix all of them.

9. The study findings were very interesting and encourage further studies to identify potential combination of markers that could help predict viral rebound dynamics in HIV cure studies with ATI. Overall, the authors did a good job on highlighting that the potential use of these signatures to guide clinical trials requires further validation, with exception of a very few remarks stating these biomarkers “can be used” to access viral remission. I suggest being careful to avoid overstatements. I would suggest changing this to “could be used”.

Authors response: We thank the reviewer for this comment, and we have adjusted this language.

10. What do the authors think would be the next steps to further validate the molecules identified in this study as biomarkers of viral remission?

Authors response: We thank the reviewer for this question and for her/his interest in the potential next steps of this study. We believe it will be important to: 1) Examine the ability of our set of plasma glycans and metabolites to reflect levels of intact and inducible HIV proviruses in tissues, the main site of HIV persistence. This analysis would demonstrate whether these factors, beyond their ability to predict time-to-viral-rebound, can also be used as non-invasive markers of the size of HIV reservoirs in tissues. Collecting tissues and analyzing intact/inducible HIV reservoirs are challenging procedures. Having plasma non-invasive biomarkers to replace these procedures can profoundly impact the design of HIV cure-oriented clinical studies. 2) Investigate the longitudinal dynamics of these markers before and after ATI. This analysis can provide unique opportunities to understand the host factors that control post-ART control of HIV. 3) Develop/test simpler methods that can be used to measure these signatures in resource-limited settings. 4) Identify the protein backbones of our glycomic signature. This identification will likely allow us a better understanding of the role of host glycosylation machinery in modulating HIV persistence. 5) Identify the type of cells responsible for the serum glycans and metabolites. 6) Investigate the potential direct and indirect functional significance of each of the key variables in our models on HIV latency reactivation and myeloid inflammation (using cells from HIV-infected ART-suppressed individuals). The potential results from this future work could open new mechanistic avenues to better understand the fundamental host biological processes, including carbohydrate metabolism, that may regulate HIV control during ART and post-ATI.

Reviewer #3

Reviewer general comment: *Giron et al's 'Non-Invasive Plasma Glycomic and Metabolic Biomarkers of Post-treatment Control of HIV' nicely written report on multiple cohorts motivates the need for non-invasive biomarkers that can predict HIV remission after ART interruption. They identified a number of glycomic and metabolic signatures associated at the univariate and leveraging a penalized methodology in combination a signature of sorts that predicts time to viral rebound and probability of viral rebound. However, I have a few questions on the statistical methodology.*

Authors response: We thank the reviewer, and we believe that we addressed all concerns raised as described below.

Specific Comments:

1. *In my reading it was not clear if the complete dataset (n=70, 4 without complete data) was used in training this algorithm from 2 data sets? If this is the case, then the AUC and c-index values reported is likely an overfit of the model and would not replicate. An alternative approach to analyzing the data would be to take the first cohort - train the model - and then test on the second cohort. Another approach could include a K-fold cross validated model where then the variance of the model estimates are reported and the prediction measures for each of the hold-out folds are reported. This would ensure a more robust modelling.*

Authors response: We thank the reviewer for this question and valuable suggestions. We apologize for the confusion. We elected not to combine the two cohorts, as 1) The first cohort (Philadelphia Cohort) was a small pilot and did not have complete data to investigate confounding effects. 2) The first cohort used the viral load (VL) > 50 copies/mL as the endpoint for the measure of time-to-viral-rebound, which is different from the second cohort (ACTG Cohort), where time to viral load (VL) ≥1000 copies/mL was used as the endpoint. This reflects the heterogeneous nature of these ATI studies in the HIV field, a point that requires standardization in the future, as highlighted by a recent review (Julg B, et al., Lancet HIV. 2019, PMID: 30885693). Thus, the two cohorts were not combined for further analysis, and only the second cohort from ACTG clinical trials with complete data (n=70) was used for variable selection to explore potential markers that could be used together in a predictive model; we have made this clearer in the Results section.

We thank the reviewer for the valuable suggestion to use a K-fold cross validated model. We agree with the Reviewer that AUC and C-index values reported from a K-fold cross-validated (CV) model would be robust; therefore, we performed this analysis.

We have now included **Supplementary Table 8** to report average AUC and C-index values and their variations from 5-fold cross-validated (CV) models, as suggested by the reviewer. These results have also been added to the corresponding section of the Results section. In summary, data using these models show a very small reduction in the prediction ability of the models (70.6% from 74% for the Cox model and 94.7% from 97.5% for the logistic model). We thank the reviewer for her/his valuable suggestion. We believe these new data add depth and validation to our findings.

Supplementary Table 8. The AUC and C-index values estimated from 5-fold cross-validated models with Lasso selected variables.

Fold for testing (n=14/fold)	Logistic model of PTC vs NC (cross-validated AUC)	Cox model for time-to-VL \geq 1000 copies/mL (cross-validated C-index)
1	0.9750	0.7111
2	0.9792	0.7386
3	0.8222	0.6813
4	0.9778	0.7045
5	0.9796	0.6923
Average (variance)	0.9468 (0.0049)	0.7056 (0.0004)

Finally, while we believe that adding another PTC cohort would have made the study stronger, PTC is an extremely scarce phenotype. In fact, to the best of my knowledge, our manuscript will be the first to include in-depth biological analysis from this large number of PTCs (27) in one study (it is one of the main strengths of our study). These 27 were identified from six clinical trials that involved 600 HIV-infected individuals who underwent ATI, and these studies were conducted in 106 clinical sites all over the U.S. in order to identify these 27 PTCs. Dr. Li recently published a survey study to identify all potential PTCs in the U.S. and Canada (Namazi G et al., J Infect Dis. 2018, PMID: 30085241) and identified only 67 PTC from many cohorts, and not all of them have biological samples stored. Thus, our study already included ~42% of all individuals identified with this phenotype in the U.S. and Canada (the most characterized of them with known clinical data and stored samples).

In summary, while PTC phenotype provides an extremely important insight into post-ART control of HIV (functional cure) and must be studied, as expected, it is rare to achieve HIV remission (a temporary functional cure from HIV). Therefore, unfortunately, it won't be possible to have an additional PTC cohort until, hopefully, in the future, it becomes more prevalent to achieve HIV remission and the number of PTCs increases.

2. My second comment is really that a C-index of 0.74 is good, but at least in many different settings is not enough for clinical improvement or change. Could the authors clarify why they believe a ~0.75 c-index is clinically meaningful?

Authors response: We are reporting two models, one that can predict time-to-viral-rebound with a moderate C-index of 0.74 and one that can differentiate PTCs from NCs with a better AUC of 0.975. We completely agree with the reviewer's comment that the C-index of 0.74 for the first model is not good enough for a predictive model. However, the purpose of our analysis was to explore potential host markers that could be used as predictors upon further investigation in future studies. Both duration and probability of HIV remission after ART-cessation are complicated virological phenomena. To the best of our knowledge, there have been no plasma biomarkers that can predict these measures. Furthermore, only a very small number of cell-based markers have been identified, over the years, to predict these clinically

important measures (and very weakly). Finding other markers that can enhance and validate our models is a needed task. That said, we think our study is a critical first step towards this goal. We have highlighted the exploratory nature of this study in multiple sites of the manuscript.

To be conservative in our exploratory results, we have now removed the old Supplementary Table 7 and only report the multivariable Cox regression analysis for time to VL \geq 1000 copies/ml. In addition, in the Results section, at the end of the paragraph starting with “*Multivariable Cox model, using Lasso technique with the cross-validation (CV), selected variables that their combination predicts time-to-viral-rebound*”, we have added the following statement: “To be conservative, a more robust estimate of the model’s performance was calculated using a 5-fold cross-validated model in ACTG cohort, which obtained the average C-index of 0.71 with a variance of 0.0004 (**Supplementary Table 8**). Together, these data suggest that the multivariable models of combined plasma glycans and metabolites markers warrant further exploration for their capacity to predict the time-to-HIV-rebound in different settings.”

REVIEWERS' COMMENTS

Reviewer #1 (Remarks to the Author):

The authors have made substantial efforts to answer my points. I have only one outstanding issue. The second link given in their rebuttal leads to an empty database. The reference code given is not recognised by Metabolomics Workbench.

Reviewer #2 (Remarks to the Author):

The reviewers have sufficiently addressed all my comments.

Reviewer #3 (Remarks to the Author):

I would like to thank the authors for this important work and for addressing my and the other reviewer comments. I feel that the comments have sufficiently been addressed and that the large number of PTCs warrant further investigation. This is a great start to further elucidation and warrants publication in Nature Communications.

Reviewer #1

Reviewer comment: The authors have made substantial efforts to answer my points. I have only one outstanding issue. The second link given in their rebuttal leads to an empty database. The reference code given is not recognised by Metabolomics Workbench.

Authors response: We thank the reviewer and apologize that he/she had problems accessing the raw data. We hypothesized that the reason is that these data are under embargo and require a username and a password to be accessed. We have now lifted the embargo on the metabolomics data on the Metabolomics Workbench. Now the data are publicly available through the following link: <https://doi.org/10.21228/m8kq59> to anyone without a username and password. Project PR001053 with two studies ST001719 with all raw data from the Philadelphia Cohort samples and ST001720 with all raw data from the ACTG Cohort. We have tested the link from several computers and ensured that the data are fully accessed.

These information and link are included in the manuscript under the Data availability statement.